# Forced expression of MSR repeat transcripts above a threshold limit breaks heterochromatin organisation

Reagan W. Ching ✉, Kalina M. Świst-Rosowska, Galina Erikson, Birgit Koschorz, Bettina Engist & Thomas Jenuwein ✉

Mouse heterochromatin is characterised by transcriptionally competent major satellite repeat (MSR) sequences and it has been proposed that MSR RNA contributes to the integrity of heterochromatin. We establish an inducible dCas9-effector system in mouse embryonic fibroblasts, where we can modulate MSR transcription through the targeting of a dCas9-Repressor or a dCas9-Activator. With this system, we can define a threshold limit of >300-fold deregulation of MSR transcript levels, above which the structural organisation of heterochromatin becomes disrupted. MEF cells expressing MSR RNA above this threshold limit are not viable and the defects in heterochromatin organisation and chromosome segregation cannot be reverted. This study highlights the importance of restricting MSR RNA output to maintain heterochromatin integrity and relates MSR transcript levels to either physiological or pathological conditions. It also reveals that the structural organisation of heterochromatin is governed by the transcriptional chromatin state and associated MSR RNA of the MSR repeats.

Heterochromatin has important functions in the structural and functional organisation of eukaryotic chromatin[1,2]. Pericentric heterochromatin is most frequently nucleated by underlying DNA repeat sequences, which in mouse cells, comprise of the A/T-rich major satellite repeat (MSR) sequences[3,4]. Various models for the establishment and maintenance of heterochromatin have been described[2,5,6]. Intriguingly, MSR sequences are not transcriptionally inert and RNA output from MSR sequences contributes to the structural organisation of heterochromatin[7] and/or the formation of phase separated RNA-chromatin condensates[8,9] and distinct compaction states[10]. MSR repeat transcripts can participate in recruiting several chromatin-modifying enzymes, such as the Suv39h methyltransferases (KMT) to heterochromatin[11–13]. MSR repeat transcripts largely remain chromatin associated and they have been proposed to form an RNA-nucleosome scaffold at heterochromatin that is supported by RNA:DNA hybrid formation[13,14].

Regulation of MSR transcription is required for early mouse embryonic development and in stabilising cellular fates[15,16]. In addition to these physiological functions, MSR expression is also relevant for pathological states. Forced expression of MSR sequences in mouse mammary glands is a tumour driver[17] and satellite repeat transcripts are up-regulated in several human cancers[18,19]. Aberrantly high levels of satellite repeat transcripts can induce transient RNA:DNA hybrid formation resulting in repeat element expansion of pericentric heterochromatin and chromosome mis-segregation[20].

Despite these important insights, the regulation and function of MSR repeat RNA is not fully understood. An early report indicated RNase A treatment of permeabilised mouse embryonic fibroblasts (MEF) lead to dispersed pericentric domains identified by an antibody recognising a branched H3K9me3 epitope[7]. However, it has been very challenging, and is probably not tolerated, to establish conditions in living cells where the RNA output from mouse heterochromatin can be modulated by the targeted expression of RNase enzymes. Alternatively, MSR targeting of dCas9-fused transcriptional activators in MEF cells[10] or in a mouse model[17] have been used to gain some insight into the dysregulation of MSR transcription.

Max Planck Institute of Immunobiology and Epigenetics (MPI-IE), Freiburg, Germany. ✉e-mail: ching@ie-freiburg.mpg.de; jenuwein@ie-freiburg.mpg.de

No cellular models for the targeting of both transcriptional repressors and transcriptional activators to MSR sequences have been described and the functional significance to restrict MSR RNA output in living cells has not been directly examined. We therefore established an inducible MSR-dCas9-effector system in MEF cells, where we can target MSR-dCas9-Repressor and MSR-dCas9-Activator to MSR repeat sequences in pericentric heterochromatin. Targeting of MSR-dCas9-Repressor enforced repression of MSR transcription and resulted in aggregation of heterochromatic foci. In contrast, targeting of MSR-dCas9-Activator to MSR sequences induced a pronounced increase in RNA output and resulted in the dispersion of DAPI-dense regions. This disruption of heterochromatin required MSR RNA to be overexpressed above a threshold limit of >300-fold deregulation. Intriguingly, targeting of MSR-dCas9-Activator to MSR induced an accumulation of chromatin marks associated with active transcription, however without removing the heterochromatic signature of the MSR-targeted chromatin regions. The forced expression of MSR transcripts also resulted in pronounced mitotic defects that are irreversible and non-viable. This study highlights the importance of restricting MSR RNA output to maintain heterochromatin integrity and also reveals that the structural organisation of heterochromatin is governed by the transcriptional chromatin state and associated MSR RNA of the underlying MSR repeat units.

## Results

### An inducible dCas9-effector system to target a transcriptional Repressor and a transcriptional Activator to mouse heterochromatin

An indication for an RNA component to contribute to the structural integrity of mouse pericentric heterochromatin was the incubation of permeabilised MEF cells with RNase A that showed dispersion of H3K9me3-enriched domains[7]. We used a doxycycline-inducible system for an MSR targeted TALE-Onconase-EGFP effector component. Onconase is an amphibian RNase A that is less susceptible to degradation by the cell intrinsic RNase inhibitor (RI)[21]. Induction of MSR-targeted TALE-Onconase-EGFP resulted in a modest reduction of MSR transcript levels and induced aggregation of heterochromatic foci (Supplementary Fig. 1).

We then considered an RNase-independent strategy that would allow the targeting of both a transcriptional Repressor and of a transcriptional Activator to MSR sequences and also define conditions that maintain a high level of cell viability. For this, we used MEF cells that display the classic focal definition of DAPI-dense heterochromatic foci[22,23]. We developed dCas9 fusions for a transcriptional Repressor (dCas9-KM)[24] and for a transcriptional Activator (dCas9-VPR)[25] that can be induced with doxycycline (dox). The system required that the inducible dCas9-effector component and the MSR guide RNA are expressed from the same plasmid (Fig. 1a). Stable inducible MEF cell lines were generated using the PiggyBac Transposon system. Due to the inducible nature of the cell system, the expression levels of the dCas9-effector components can be equalised by varying induction times and/or dox concentrations. The induction conditions, which resulted in comparable dCas9 protein levels as observed by Western blot analysis, were: 24 h for the MSR-dCas9-Control (10 ng/ml dox), 24 h for the MSR-dCas9-Repressor (100 ng/ml dox) and 6 h for the MSR-dCas9-Activator (10 ng/ml dox) (Fig. 1b).

To determine if the MSR-dCas9-effector components were targeted to pericentric heterochromatin, cells were prepared for immunofluorescence microscopy and stained for dCas9 protein and DAPI. Due to the A/T-richness of MSR DNA[3,4], MSR-containing pericentric regions can be visualised as DAPI-dense regions. Under our optimised induction conditions, the MSR-dCas9-effector components localised to these DAPI-dense regions (Fig. 1c). A 24 h induction of MSR-dCas9-Control and of the MSR-dCas9-Repressor or a 6 h induction of MSR-dCas9-Activator does not impair cell viability (Supplementary Fig. 2a)

and does not significantly alter the cell cycle (Supplementary Fig. 2b) when compared to uninduced MEF cells. Equivalent induction times were not possible between the three cell lines due to massive cell death observed when inducing the MSR-dCas9-Activator for 24 h (Supplementary Fig. 2a).

### Targeting of inducible MSR-dCas9-effector components modulates RNA output from heterochromatin

After the targeting of the MSR-dCas9-effector components to pericentric heterochromatin was confirmed, we wished to determine how effective they were in modulating MSR transcription. Total RNA was purified from uninduced and induced cells and RT-qPCR was performed with MSR specific primers. An ~4000-fold increase in MSR RNA was observed upon induction of the MSR-dCas9-Activator and an 80% decrease in MSR transcripts was detected for the induced MSR-dCas9-Repressor (Fig. 1d). In addition, we also observed an ~100-fold increase in minor satellite expression (with the induced MSR-dCas9-Activator), but no significant decrease for minor satellite transcripts in the induced MSR-dCas9-Repressor cell line. We did not detect any changes in the expression of LINE-1 (L1Md_A) repeat transcripts in any of the cell lines tested (Fig. 1d). To verify these results, RNA-seq was performed using ribo-depleted total RNA, and differential expression analysis was performed with induced MSR-dCas9-Activator and induced MSR-dCas9-Repressor samples relative to induced MSR-dCas9-Control, and also compared between uninduced and induced MSR-dCas9-Control. As shown in the MA-plots, MSR transcripts (GSAT_MM) and minor satellite transcripts (SYNREP_MM) were significantly upregulated in the induced MSR-dCas9-Activator cell line (Fig. 1e). In the induced MSR-dCas9-Repressor cell line, MSR transcripts were further repressed. The transcription of LINE-1 (L1Md_A) repeats was not altered in the presence of either MSR-dCas9-Activator or MSR-dCas9-Repressor (Fig. 1e).

The differential RNA-seq analysis in the uninduced and induced MSR-dCas9-Activator cell lines also indicated partial dysregulation (both up-regulation and down-regulation) of other repeat transcripts, such as ERV retrotransposons, tRNA repeats, LINE elements and other repeats (Fig. 1e). The genomic organisation of MSR repeat arrays is very complex and in addition to the MSR repeat units is interspersed with sequences of some of these other repeat classes[26]. The detection of RNA reads for these other repeat classes could probably be an indirect effect of the overexpression of MSR RNA and may be related to the generation of read-through or chimeric transcripts. This could, at least in part, also explain the detection of RNA reads for minor satellite repeat sequences, which are in the immediate vicinity of the pericentric MSR repeat arrays. To address this question, long-read RNA sequencing would be required. In sum, we conclude that our inducible MSR-dCas9-effector components can effectively modulate MSR RNA output.

### Perturbed heterochromatin organisation in MEF cells expressing MSR-dCas9-effector components

We observed a considerable disruption of DAPI-positive heterochromatin regions in the MSR-dCas9-Activator cell line. To obtain increased contrast and to better quantify the changes in the organisation of heterochromatin, we next performed DNA-FISH with MSR-specific probes using Airyscan confocal microscopy (Fig. 2a). By collecting optical z-stacks and creating a 3D mask of the MSR-FISH signal, we were able to measure differences in the sphericity and volume of heterochromatin. Sphericity was chosen as a measurement of heterochromatin dispersion, since heterochromatin is normally organised in rounded focal structures (DAPI-dense foci). Only the expression of the MSR-dCas9-Activator resulted in a significant change in the sphericity of heterochromatin, as it was quantified by the MSR DNA-FISH signal. The sphericity of heterochromatin was not significantly altered with the induced MSR-dCas9-Repressor (Fig. 2a, right panels). When comparing the overall volume of heterochromatin regions, both

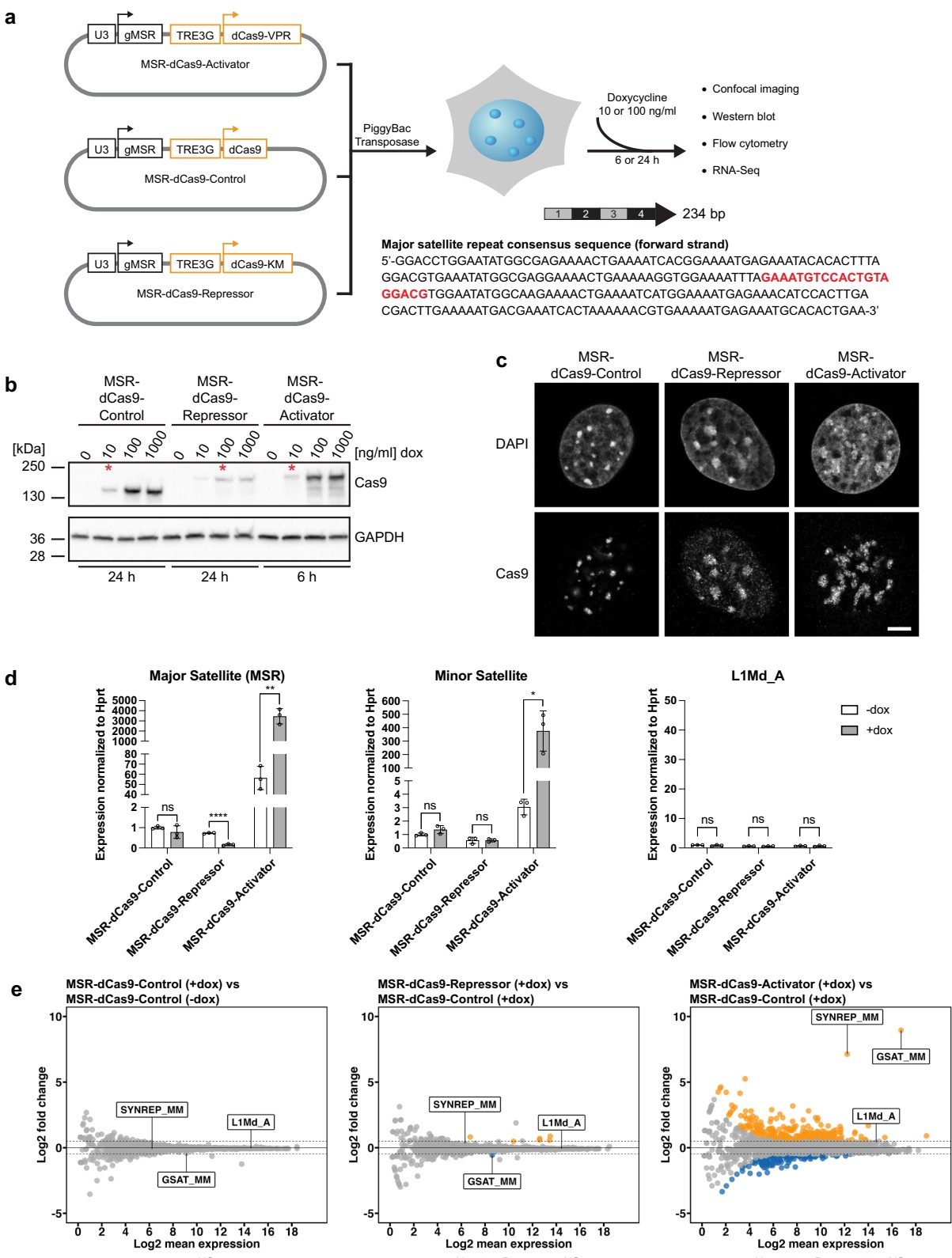

the expression of the MSR-dCas9-Activator and of the MSR-dCas9-Repressor resulted in a considerable increase of volume of DAPI- and MSR-positive areas (Fig. 2a, right panels). No apparent changes in sphericity or volume of heterochromatin were detected in the MSR-dCas9-Control cells.

With the expression of the MSR-dCas9-Repressor, heterochromatin regions did increase in volume. This could result from the

clustering of smaller DAPI-dense foci. To test this, we counted the number of centromeres per DAPI-dense region. Centromeres are normally located at the periphery of DAPI-dense foci and can be visualised by staining with a CREST antiserum which recognises the centromeric proteins CENP-A, CENP-B, and CENP-C[27]. We observed that the number of CREST puncta per DAPI-dense region was largely unchanged when expressing either MSR-dCas9-Control or MSR-dCas9-

**Fig. 1 | Targeting of inducible MSR-dCas9-effector components to modulate RNA output from heterochromatin. a** Diagram of inducible MSR-dCas9-effector components: MSR-dCas9-Control, MSR-dCas9-Activator (VPR) and MSR-dCas9-Repressor (KM). Also shown is the DNA sequence of a single unit (234 bp) of the MSR consensus, with the sequence of the dCas9 guide RNA highlighted in bold red. **b** Western blot analysis for the detection of MSR-dCas9-Control, MSR-dCas9-Repressor and MSR-dCas9-Activator after doxycycline (dox) induction. GAPDH expression is shown as a loading control. Induction conditions that result in comparable expression of MSR-dCas9-Control, MSR-dCas9-Repressor and MSR-dCas9-Activator used in this study are indicated with red asterisks. **c** Immunofluorescence for the localisation of MSR-dCas9-Control, MSR-dCas9-Repressor and MSR-dCas9-Activator. MEF cells were counterstained with DAPI. Scale bar is 5 μm. Images were acquired from three independent experiments. **d** RT-qPCR analysis for major

satellite repeat (MSR), minor satellite repeat and LINE L1Md_A repeat transcripts in inducible MSR-dCas9-Control, MSR-dCas9-Repressor and MSR-dCas9-Activator MEF cells. Expression is normalised to *Hprt* and is relative to MSR-dCas9-Control (-dox) (mean±SD). $n = 3$ independent experiments. Asterisks indicate statistically significant differences (*, $p \leq 0.05$, **, $p \leq 0.001$, ****, $p \leq 0.0001$, ns, not significant, two-sided multiple *t* tests). **e** MA-plots depicting differential repeat element expression in inducible MSR-dCas9-Control, MSR-dCas9-Repressor and MSR-dCas9-Activator MEF cells, as determined by RNA sequencing (RNA-seq). Major satellite repeats (GSAT_MM), minor satellite repeats (SYNREP_MM) and LINE-1 (L1Md_A) repeats are marked on the plots. Up-regulated repeats are represented by orange dots and down-regulated repeats by blue dots. Grey dots represent not statistically significant changes. $n = 3$ independent experiments.

Activator. Expressing the MSR-dCas9-Repressor, however, resulted in an altered ratio of CREST puncta per DAPI-dense region, with some DAPI-dense regions containing an increased number of CREST puncta (see violin plots in Supplementary Fig. 3a).

Several chromatin factors are core components of mouse pericentric heterochromatin, such as the HP1 family of proteins and members of the high mobility group of proteins (e.g. HMGA1). The HP1 proteins recognise H3K9me3 via their chromodomain and the A/T-hook domains of HMGA1 bind to A/T-rich DNA sequences. Both H3K9me3 and A/T-richness are found at pericentric heterochromatin[28–31]. The altered heterochromatin organisation could be the result of impaired localisation of these proteins to dispersed DAPI-dense regions. To address this, we analysed the distribution of HP1α (Fig. 2b) and of HP1β, HP1γ and HMGA1 (Supplementary Fig. 4) in uninduced and induced MSR-dCas9-effector cells. We observed that these heterochromatic proteins were still localised to DAPI-dense regions, regardless of the distorted organisation of heterochromatin with either induced MSR-dCas9-Repressor or MSR-dCas9-Activator. To visualise the sub-nuclear localisation and overlap of HP1 proteins and of HMGA1 with heterochromatin, we used linescans (see Methods) that connect DAPI-positive regions with DAPI-negative areas (Fig. 2c, Supplementary Fig. 4).

Taken together, the data indicate that targeting of MSR-dCas9-effector components leads to the disorganisation of heterochromatin. Targeting an MSR-dCas9-Repressor induces a volume increase due to heterochromatin aggregation, while targeting an MSR-dCas9-Activator results in heterochromatin disruption, as indicated by a decrease in sphericity of DAPI-dense regions. Both, the aggregation and disruption of heterochromatin occur without apparent changes in the localisation of known core components of DAPI-dense heterochromatin.

## RNA polymerase II and histone acetylation redistribute and concentrate at MSR-targeted dCas9-Activator domains

We next addressed whether changes in histone modifications may correlate with either the aggregation or disruption of heterochromatin. We applied indirect immunofluorescence to stain for repressive and activating histone H3 marks and co-localisation with the dCas9-effector components and DAPI-dense regions. For the detection of H3K9me3 (constitutive heterochromatin), we used a canonical H3K9me3 antibody (see "Methods"), as the antibody recognising branched H3K9me3 epitopes[7] was no longer available. H3K9me3 signals were largely unaltered in the induced MSR-dCas9-Control and MSR-dCas9-Repressor cells and localised to the classic DAPI-dense foci or to clustered DAPI-dense regions (Supplementary Fig. 3b, left panel). In the induced MSR-dCas9-Activator cells, the H3K9me3 signals persist to overlap with the perturbed DAPI-dense regions (Supplementary Fig. 3b, left panel). This may explain why the HP1 proteins also remained at the distorted DAPI-dense regions, even in the presence of high transcriptional activity. For H3K27me3 (facultative heterochromatin), we did not observe enrichment over DAPI-dense regions in MSR-dCas9-Control and MSR-dCas9-Repressor cells. Although there is a global increase for H3K27me3 in the MSR-dCas9-Activator cell line,

H3K27me3 signals are largely excluded from DAPI-dense regions (Supplementary Fig. 3b, right panel). The increase in H3K27me3 does not appear to be caused by the expression of the MSR-dCas9-Activator, since bulk H3K27me3 levels was equal between uninduced and induced MSR-dCas9-Activator state (Supplementary Fig. 5).

We then analysed the activating histone marks H3K4me3, found over promoters, and H3K36me3, found over active gene bodies[32]. We observed no accumulation of these histone marks over heterochromatin, nor co-localisation with any of the MSR-dCas9-effector components. Even expression of the MSR-dCas9-Activator did not result in an enrichment of H3K4me3 or H3K36me3 at DAPI-dense regions (Supplementary Fig. 3c). When probing for chromatin marks associated with active transcription, such as initiating RNA polymerase II (RNAPII) (Ser5phos) and pan-acetylated histone H3, we observed a typical distribution pattern in the MSR-dCas9-Control and MSR-dCas9-Repressor MEF cells; i.e. these marks were nuclear diffuse and excluded from DAPI-dense regions (Fig. 2c). When probing in the MSR-dCas9-Activator MEF cells, we observed a redistribution of RNAPII (Ser5phos) from nuclear diffuse to DAPI-dense regions (Fig. 2c, left panel). This was accompanied by a reallocation of pan-acetylated H3 (H3 panAc) (Fig. 2c, right panel), which mirrored the pattern observed for RNAPII (Ser5phos). This reallocation occurred without an increase in bulk levels of H3 panAc (Supplementary Fig. 5). The redistribution of RNAPII and the associated histone hyperacetylation is consistent with an earlier study that also used targeting of a dCas9-Activator to heterochromatin in MEF cells[10].

## Upregulated MSR repeat transcripts remain associated with MSR-targeted dCas9-Activator domains and form RNA: DNA hybrids

Transcription at MSR DNA occurs bi-directionally producing both forward (purine-rich) and reverse (pyrimidine-rich) RNA strands[13,33]. With our inducible MSR-dCas9-effector components, we wanted to determine the localisation of MSR RNA relative to DAPI-dense regions. We performed immunoRNA-FISH using strand specific MSR probes and counterstained for Cas9 protein and DAPI. The maximum exposure time was determined with the MSR-dCas9-Activator cells, and all RNA-FISH signals between induced cell lines are relative to the expressed MSR-dCas9-Activator. In the induced MSR-dCas9-Control cells, only a very faint signal for MSR RNA was detected with either the antisense (recognising forward transcripts) or sense (recognising reverse transcripts) probe (Fig. 3a). No MSR RNA-FISH signal was detectable for the induced MSR-dCas9-Repressor cells. In the induced MSR-dCas9-Activator cells, both the forward and the reverse MSR transcripts were drastically increased, with a preference for production of the MSR forward transcript. In addition, the MSR transcripts were centred in and around DAPI-dense regions and colocalising with the MSR-dCas9-Activator domains (Fig. 3a). RNase A treatment after fixation but prior to probe hybridisation was performed as a control to ensure that the FISH signal was due to hybridisation to RNA and not to DNA (Fig. 3a).

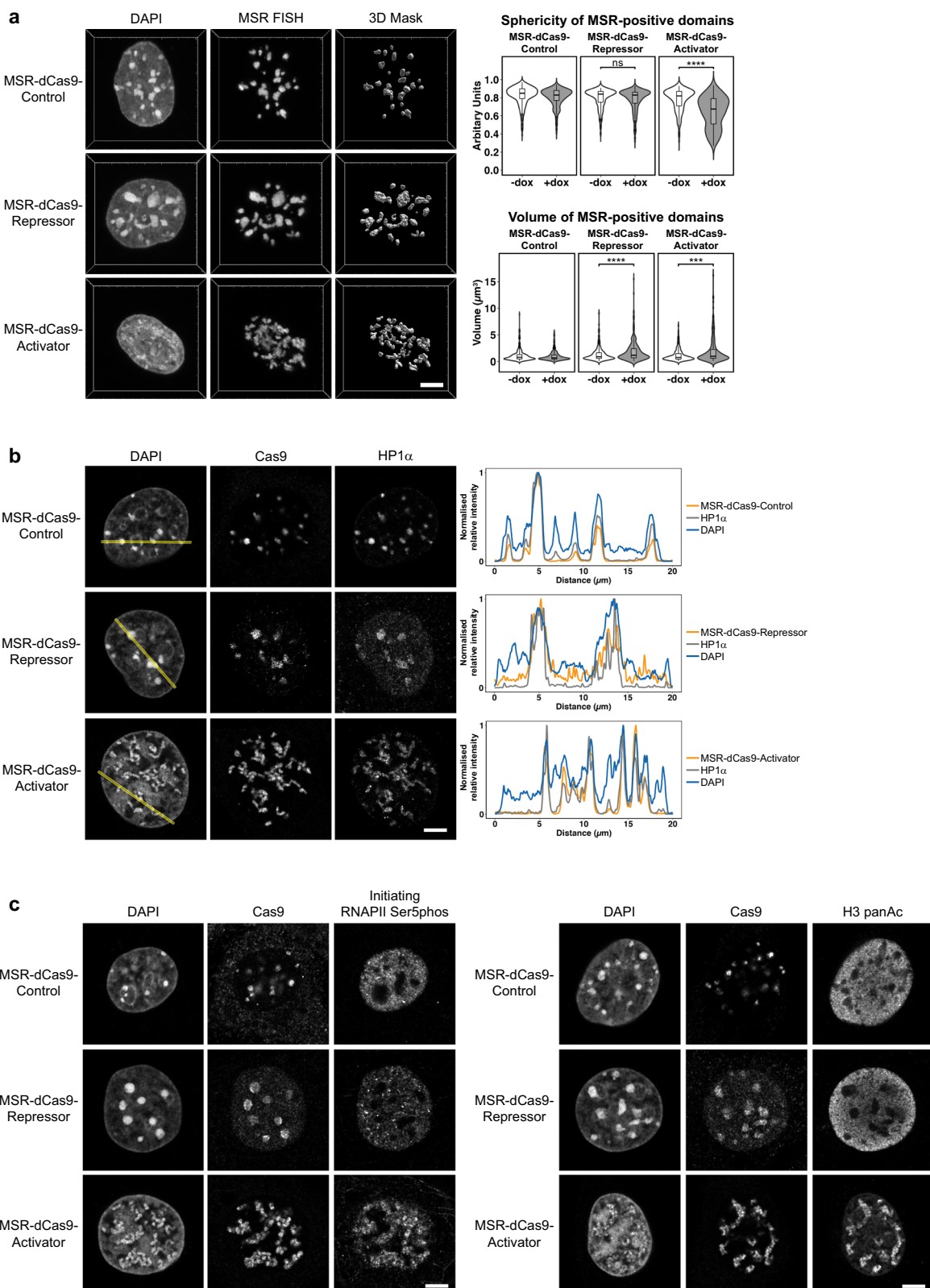

MSR RNA is chromatin-associated and prone to form RNA:DNA hybrids[13,14]. It is thus possible that some of the overexpressed MSR RNA in the induced MSR-dCas9-Activator MEF cells could also exist as RNA:DNA hybrids. To address this, an RNA:DNA dot blot probed with the S9.6 antibody (which detects RNA:DNA hybrids) was performed and quantified. We observed a two-fold increase in global RNA:DNA hybrid formation by comparing nucleic acid samples from uninduced

and induced MSR-dCas9-Activator MEF cells (Fig. 3b). To directly examine whether RNA:DNA hybrids are preferably formed by MSR sequences, we next used RNA:DNA immunoprecipitation (RDIP) with the S9.6 antibody on purified genomic DNA and quantified enriched material by qPCR with repeat-specific primer pairs. This RDIP analysis showed that RNA:DNA hybrids are significantly increased for MSR sequences only in samples from the induced MSR-dCas9-Activator

**Fig. 2 | Perturbed heterochromatin organisation in MEF cells expressing MSR-dCas9-effector components. a** Immuno-DNA FISH analysis for MSR sequences of inducible MSR-dCas9-Control, MSR-dCas9-Repressor and MSR-dCas9-Activator MEF cells. In addition to the DNA-FISH signal, 3D masks were created to quantify changes in sphericity and volume of MSR-positive domains (right panels). MSR-positive domains analysed: MSR-dCas9-Control n = 509 (−dox), n = 469 (+dox); MSR-dCas9-Repressor n = 404 (−dox), n = 371 (+dox); MSR-dCas9-Activator n = 344 (−dox), n = 254 (+dox). Box plots shown are centred around the median value with an interquartile range (IQR) defined by the 1st and 3rd quartiles and whiskers extending to 1.5xIQR. Outliers are plotted as individual points. Asterisks indicate statistically significant differences (***, $p \leq 0.001$, ****, $p \leq 0.0001$, two-sided unpaired Mann–Whitney test). Scale bar is 5 μm. **b** Double immunofluorescence in inducible MSR-dCas9-Control, MSR-dCas9-Repressor and MSR-dCas9-Activator MEF cells for localisation of dCas9-effector components and HP1α. Nuclei were counterstained with DAPI. Linescans were used to visualise the colocalisation of DAPI (blue), Cas9 (orange), and HP1α (grey) signals as shown in the panels on the right. For each sample n ≥ 30 cells were analysed. Scale bar is 5 μm. **c** Double immunofluorescence in inducible MSR-dCas9-Control, MSR-dCas9-Repressor and MSR-dCas9-Activator MEF cells for localisation of MSR-dCas9-effector components and initiating RNAPII (Ser5phos) (left panel) or for MSR-dCas9-effector components and H3 panAc (right panel). Nuclei were counterstained with DAPI. For each sample n ≥ 30 cells were analysed. Scale bar is 5 μm.

MEF cells (Fig. 3c). Treatment of the samples with RNase H prior to RDIP decreased these RNA:DNA hybrid signals. While some RNA:DNA hybrids were also found for minor satellite repeat sequences, no RNA:DNA hybrids were detected for LINE-1 (L1Md_A) repeats. Thus, a portion of the overexpressed MSR RNA in the induced MSR-dCas9-Activator MEF cells can exist as RNA:DNA hybrids which could facilitate chromatin association of MSR repeat transcripts and restrict their localisation to the vicinity of MSR DNA repeat arrays within or around DAPI-dense regions.

## Heterochromatin disruption requires a threshold limit for MSR transcript deregulation

The levels of MSR RNA vary during distinct physiological and pathological states. For example, during mouse embryogenesis at the 2-cell stage embryo, there is a burst of MSR RNA of around 50–120-fold upregulation as analysed by MSR strand-specific qPCR[16]. This increase in MSR transcripts has been proposed to assist in chromocenter formation[16]. In contrast, MSR transcript levels can reach very high abundance in *Brca1*-deficient mouse mammary tumours[34] and satellite repeats are often aberrantly deregulated in human cancer where they can cause aneuploidies[17–19]. We wished to determine if a threshold limit for MSR overexpression is required for heterochromatin disruption. For this, we attenuated expression of the MSR-dCas9-Activator with low dose titrations of doxycycline from 10 ng/ml to 0.5 ng/ml for 6 h induction times. Although the levels of MSR-dCas9-Activator protein become barely visible below a doxycycline concentration of 4 ng/ml (Fig. 4a), we observe a significant increase (>300–800-fold) in MSR transcript levels starting with doxycycline concentrations between 2 and 3 ng/ml (Fig. 4b). MSR transcript levels progressively increased and plateaued with a > 4000-fold overexpression at a doxycycline concentration of 10 ng/ml. Importantly, disruption of heterochromatin becomes apparent with a ~ 300–800-fold MSR overexpression, where around 29–62% of the induced MSR-dCas9-Activator MEF cells show dispersed DAPI-dense regions (Fig. 4c). Thus, disruption of heterochromatin occurs at a threshold limit of MSR overexpression that is defined by a > 300-fold deregulation of MSR transcripts, as gauged by RT-qPCR (Fig. 4b, red highlighted segment).

The above analysis uses RT-qPCR to quantify differences in MSR transcript levels. RT-qPCR is likely to overestimate the relative abundance of repeat transcripts since there are multiple target sites for PCR primers in multi-copy repeat RNA. We therefore also interrogated normalised read counts for MSR sequences from our HiSeq RNA sequencing data which allows for a comparative analysis across distinct genetic backgrounds and different sequencing depths (see "Methods"). A proportional conversion of RT-qPCR values corresponding to the MSR-dCas9-Activator (dox 0 ng/ml and dox 10 ng/ml) was performed to calculate a threshold limit also for normalised MSR RNA-seq counts (see Methods). With this, a threshold limit of 13,000–37,000 normalised RNA-seq counts could be defined. In uninduced MSR-dCas9-Control MEF cells, around 600 normalised reads for MSR sequences are found and this is threefold below the averaged normalised reads for the housekeeping gene *Hprt*. The number of normalised reads for MSR sequences is elevated in uninduced MSR-dCas9-Activator (>2000 normalised reads) but this is below the threshold limit. This increases to >220,000 normalised reads in the fully induced (dox 10 ng/ml) MSR-dCas9-Activator MEF cells which is above the threshold limit (Fig. 4d). We then compared normalised reads for MSR sequences in wt and *Suv39h* double-null MEF cells and in 6KO MEF cells which lack all six genes encoding H3K9 KMT enzymes[35]. In wt MEF cells, a base level of around 250 normalised reads for MSR sequences is found and this number is increased to >1000 normalised reads in *Suv39h* double-null MEF cells and to >20,000 normalised reads in the 6KO MEF cells (Fig. 4d). Notably, *Suv39h* double-null MEF cells maintain the structural organisation of DAPI-dense foci[35,36] (below the threshold), whereas 6KO MEF cells completely collapse heterochromatin[35] (above the threshold). Finally, we analysed available datasets for a mouse tumour model that described the forced overexpression of MSR satellite RNA as a tumour driver in mouse mammary glands[17]. In both the mammary gland (>50,000 normalised reads) and mammary tumour (350,000 normalised reads) samples, normalised reads were above the threshold limit (Fig. 4d). The organisation of heterochromatin in normal vs tumour mammary glands was not analysed[17].

## RNA transcription and hyperacetylated chromatin drive heterochromatin disruption

Targeting of the MSR-dCas9-Activator to heterochromatin results in the redistribution and accumulation of RNAPII at DAPI-dense regions, together with inducing high levels of histone H3 acetylation (see Fig. 2c). It thus remained unclear whether the MSR-dCas9-Activator mediated disruption of heterochromatin is caused by excessive RNA output and/or an altered chromatin state. To address this question, we engaged in three experimental approaches. First, we optimised protocols for RNase A digestion of permeabilised MEF cells that are embedded in low melting point agarose (see "Methods"). We then performed a time course to identify induction conditions at which DAPI-dense regions start to become dispersed. We observe that at 3–4 h of induction of MSR-dCas9-Activator (using a doxycycline concentration of 10 ng/ml) (Supplementary Fig. 6a), MSR transcript levels are significantly increased (Supplementary Fig. 6b) and 46% (3 h time point) to 95% (4 h time point) of the cells display disrupted heterochromatin (Supplementary Fig. 6c). MSR-dCas9-Activator MEF cells were embedded in low-melting point agarose, induced for either 3 h or 4 h, permeabilised, and then incubated with RNase A. This RNase A treatment resulted in loss of HP1α association with dispersed DAPI-dense regions but did not revert disrupted heterochromatin to focal heterochromatin (Supplementary Fig. 6d). As a control, RNA-dependent localisation of nucleophosmin (NPM1) at the nucleoli was lost by RNase A digestion.

Second, we used inhibition of RNAPII with DRB. Since adding DRB simultaneously with doxycycline will block the generation of MSR-dCas9-Activator mRNA, we allowed for 1 h of induction (using a doxycycline concentration of 10 ng/ml) and then a 5 h incubation of DRB alone. This treatment did not considerably attenuate the production of MSR-dCas9-Activator (Fig. 5a), but did significantly decrease MSR RNA output (Fig. 5b). Although the MSR transcript levels were reduced

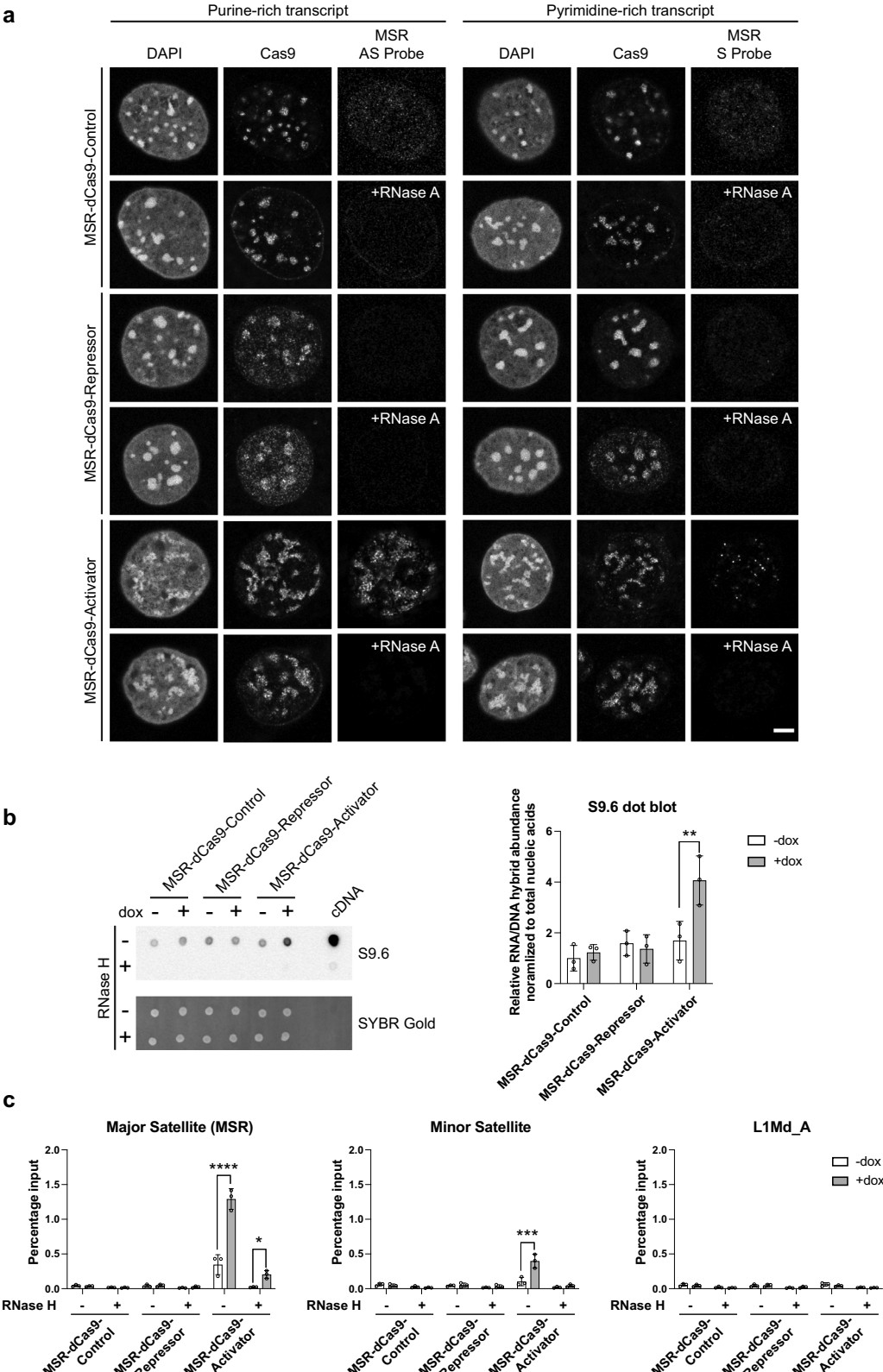

below the threshold limit, we only observed a very modest regression in the fraction of cells (88–81%) that display dispersed DAPI-dense regions (Fig. 5c).

Third, we were altering the chromatin state by using an inhibitor (A-485), which blocks the CBP and p300 HAT enzymes, that greatly reduces histone H3 acetylation[37]. When MSR-dCas9-Activator MEF cells were induced in the presence of increasing A-485 concentrations

(0–10 μM), MSR-dCas9-Activator levels remained high, but the levels of panAc H3 significantly decreased (Fig. 5d). Importantly, increasing the concentration of A-485 reduced the fraction of induced MSR-dCas9-Activator MEF cells with dispersed DAPI-dense regions and prevented the disruption of heterochromatin. At 5 μM A-485, around 70% of induced MSR-dCas9-Activator MEF cells show undisrupted (focal) heterochromatin and greatly reduced, nuclear diffuse signals

**Fig. 3 | Upregulated MSR repeat transcripts remain associated with MSR-targeted dCas9-Activator domains and form RNA:DNA hybrids. a** Immuno-RNA FISH in inducible MSR-dCas9-Control, MSR-dCas9-Repressor and MSR-dCas9-Activator MEF cells for localisation of MSR-dCas9-effector components and forward (purine-rich) MSR repeat transcripts (left panel) or for MSR-dCas9-effector components and reverse (pyrimidine-rich) MSR repeat transcripts (right panel). Nuclei were counterstained with DAPI. In parallel, cells were incubated with RNase A ( + RNase A) prior to MSR probe hybridisation. For each sample $n \geq 60$ cells were analysed. Scale bar is 5 µm. **b** RNA:DNA hybrid dot blot with the S9.6 antibody on nucleic acid samples from MSR-dCas9-Control, MSR-dCas9-Repressor and MSR-dCas9-Activator MEF cells uninduced or induced with doxycycline (dox). Samples were also treated with or without RNase H to determine RNase H sensitivity of the immunoprecipitated nucleic acids. The membrane was then stained with SYBR Gold to quantify total nucleic acids. As a control for RNase H activity, a cDNA sample was included. The histogram on the right is the quantification of the S9.6 signal from the dot blot normalised to the SYBR Gold signal (mean ± SD). $n = 3$ independent experiments. Asterisks indicate statistically significant differences (**, $p \leq 0.0019$, two-way ANOVA, Šídák's test). **c** RNA:DNA hybrid immunoprecipitation (RDIP) with the S9.6 antibody on genomic DNA from MSR-dCas9-Control, MSR-dCas9-Repressor and MSR-dCas9-Activator MEF cells uninduced or induced with doxycycline (dox). Samples were treated with or without RNase H to determine RNase H sensitivity of the immunoprecipitated nucleic acids. RNA: DNA hybrids were quantified by qPCR using primers against MSR, minor satellite repeats and LINE-1 (L1Md_A) repeats. For each histogram, values were normalised relative to MSR-dCas9-Control (−dox, −RNase H) (mean ± SD). $n = 3$ independent experiments. Asterisks indicate statistically significant differences (*, $p \leq 0.0102$, ***, $p \leq 0.001$, ****, $p \leq 0.0001$, two-way ANOVA, Šídák's test).

for H3 panAc that no longer overlapped with the targeted MSR-dCas9-Activator (Fig. 5f). However, under these conditions, MSR RNA was also significantly down-regulated to MSR transcript levels below the threshold limit (Fig. 5e). We then combined both CBP/p300 HAT and RNAPII inhibition. When induced MSR-dCas9-Activator MEF cells were treated with 5 µM A-485 and 300 µM DRB, the fraction of induced MSR-dCas9-Activator MEF cells with undisrupted heterochromatin was significantly higher (57% of cells) than when using A-485 alone (37% of cells) (Fig. 5g). We conclude from these data that the disruption of heterochromatin is driven by both histone H3 hyperacetylation and highly elevated transcriptional activity at DAPI-dense regions. In addition, it appears that RNA output and an altered chromatin state cannot be uncoupled.

### Delayed mitotic progression and chromosome mis-segregation in MEF cells expressing MSR-dCas9-Activator

Forced expression of MSR RNA resulted in an increase in RNA:DNA hybrid formation at heterochromatic regions (Fig. 3b, c). Accumulation of RNA:DNA hybrids has been associated with genomic instability[38] and may manifest itself as mitotic defects. To address this, we induced the MSR-dCas9-effector components in the presence of RO-3306, a CDK1 inhibitor, which arrests cells in the G2 stage of the cell cycle[39]. After a 24 h incubation with RO-3306, the inhibitor was washed out and cells were released to progress through mitosis. Cell samples were collected every 30 min, for an interval of up to 2.5 h, and analysed by western blot and immunofluorescence. The MSR-dCas9-Control, as well as the MSR-dCas9-Repressor and the MSR-dCas9-Activator were expressed at comparable levels throughout mitosis (Fig. 6a). Exit from G2 and progression through mitosis was verified with the mitotic histone mark H3S10phos. While H3S10phos levels reached a maximum within 1 h after RO-3306 removal for all three cell lines, the bulk H3S10phos signals are reduced in the MSR-dCas9-Activator samples. This could suggest that there are less mitotic cells in the MSR-dCas9-Activator cell population. We quantified the percentage of H3S10ph-positive cells at the 30 min intervals post RO-3306 removal. We observed that at each 30 min interval, cells induced to express MSR-dCas9-Activator had a lower percentage of H3S10ph-positive cells as compared to cells expressing either MSR-dCas9-Control or MSR-dCas9-Repressor (Fig. 6b). This indicates induced MSR-dCas9-Activator MEF cells have a delayed progression through mitosis.

We next analysed chromosome segregation in metaphase, anaphase and telophase stages of mitosis. While chromosome segregation appeared normal in both uninduced and induced MSR-dCas9-Control and MSR-dCas9-Repressor cells, induced MSR-dCas9-Activator cells displayed pronounced mitotic defects. This was most prominent in anaphase that showed a high frequency of lagging chromosomes, chromosome bridges and multiple mitotic spindles (Fig. 6c). Chromosome defects further accumulate in telophase, where abscission is

impeded (Fig. 6c). These mitotic defects are specific for the expression of the MSR-dCas9-Activator, as they were not observed for uninduced or induced MSR-dCas9-Control or MSR-dCas9-Repressor cells (Supplementary Fig. 7).

### Induced expression of MSR-dCas9-activator irreversibly compromises cell viability

The above data show that RO-3306 synchronised and released cells that were induced to express MSR-dCas9-Activator exhibit severe mitotic chromosome defects. We also observed that unsynchronised cells expressing MSR-dCas9-Activator accumulate a high percentage of apoptotic and dead cells, if they were induced for 24 h (Supplementary Fig. 2a). To address whether apoptosis and cell death in cells expressing MSR-targeted dCas9-effector components would be dependent or independent of progression through mitosis, we compared induced cell samples that were either unsynchronised or RO-3306 synchronised and also RO-3306 synchronised and released or RO-3306 synchronised and blocked. Following induction, doxycycline was washed out and percentages of apoptotic and dead cells were quantified by flow cytometry viability staining either 0 h or 24 h after dox removal (Fig. 7a). No decrease in cell viability was apparent for any of the three cell lines expressing MSR-dCas9-Control, MSR-dCas9-Repressor, or MSR-dCas9-Activator at 0 h after dox removal (Fig. 7a, left panel). At 24 h after doxycycline removal, cells induced to express MSR-dCas9-Control maintained high cell viability; cells induced to express MSR-dCas9-Repressor presented a low percentage of apoptotic and dead cells, if they were RO-3306 blocked; by contrast, the major proportion of cells that were induced to express MSR-dCas9-Activator displayed apoptosis and cell death, regardless if they were unsynchronised, RO-3306 synchronised and released or RO-3306 synchronised and blocked (Fig. 7a, right panel). RO-3306 blocked cells are arrested in the G2 stage of the cell cycle and cannot enter mitosis. Thus, accumulation of mitotic defects described above does not appear to be the only cause for the severely compromised viability of cells induced to express MSR-dCas9-Activator.

To investigate this further, we were next examining gene expression pathways that may be altered and could possibly underlie the compromised viability of MEF cells overexpressing MSR RNA. For all three MSR-dCas9-effector cell lines, we generated HiSeq total RNA libraries (see "Methods") and compared the uninduced with the induced state and also with recovery phases at 3, 6, 12, and 24 h after doxycycline removal. MSR-dCas9-effector components progressively decline in doxycycline-free media and their protein levels are barely detectable at 12 h and 24 h post-induction (Supplementary Fig. 8a). Accordingly, during the 24 h recovery phases, MSR transcript levels are either largely unchanged (for the MSR-dCas9-Control) or are restored (for the MSR-dCas9-Repressor) to MSR transcript levels observed in the uninduced state. In contrast, however, forced MSR transcript levels in the MSR-dCas9-Activator MEF cells remain in high abundance and

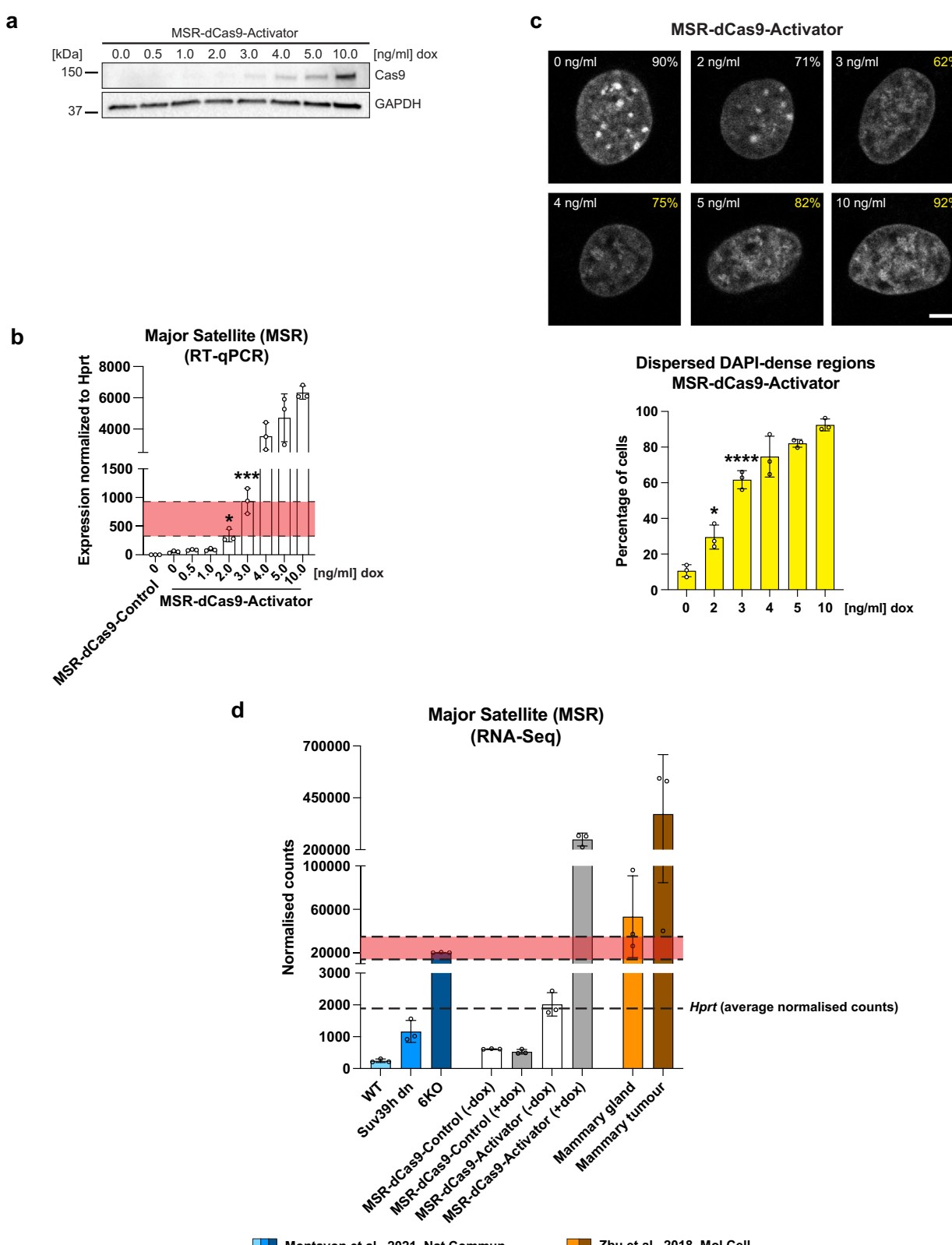

do not decline, even at 24 h post-induction as shown with RT-qPCR analysis (Supplementary Fig. 8b).

Heatmaps with k-means clustering indicated that differentially expressed genes between MSR-dCas9-Control and MSR-dCas9-Activator during the recovery phases post-induction group into three clusters (Fig. 7b). Gene Ontology (GO) analysis on these three clusters revealed that pathways involved in energy production (cluster 1, up-regulated) or signalling and development (cluster 3, down-regulated) were altered (Supplementary Fig. 8c). Importantly, pathways associated with ncRNA processing, DNA break repair and cell cycle progression and mitosis are progressively down-regulated in the MSR-dCas9-Activator post-induction (cluster 2) (Fig. 7c). This strongly suggests that cells with persistent overexpression of MSR RNA will offset DNA repair and chromosome segregation control and accede

**Fig. 4 | Heterochromatin disruption requires a threshold limit for MSR transcript deregulation. a** Western blot analysis for the detection of MSR-dCas9-Activator after induction with increasing concentrations of doxycycline (dox). GAPDH expression is shown as a loading control. **b** RT-qPCR analysis for MSR transcripts from MSR-dCas9-Activator MEF cells induced with increasing concentrations of doxycycline (dox). Expression is normalised to *Hprt* and is relative to MSR-dCas9-Control (-dox) (mean±SD). *n* = 3 independent experiments. The red segment indicates the threshold range of MSR transcript deregulation (between 300–800-fold), within which heterochromatin starts to become disrupted. (*, *p* ≤ 0.0481, ***, *p* ≤ 0.0003, one-way ANOVA, Dunnett's test). *n* = 3 independent experiments. **c** Confocal imaging of DAPI-dense regions (DAPI counterstaining) in MSR-dCas9-Activator MEF cells induced with increasing concentrations of doxycycline (dox). The percentages reflect the fraction of cells with either undispersed (white) or dispersed (yellow) DAPI-dense regions. (*, *p* ≤ 0.0222, ****, *p* ≤ 0.0001, one-way ANOVA, Dunnett's test). Scale bar is 5 µm. For each condition *n* ≥ 240 cells were analysed from *n* = 3 independent experiments. Quantification of the imaging data is shown in the bar graph below (mean ± SD). **d** Bar graph of normalised counts from HiSeq RNA sequencing comparing MSR expression in *Suv39h dn* and H3K9 KMT 6KO MEF cells (Montavon et al. 2021) (blue), in uninduced and induced MSR-dCas9-Activator MEF cells (this study) (white and grey) and in mouse mammary tumours (Zhu et al. 2018) (orange) (mean ± SD). Fold increase of MSR RNA in the *Suv39h dn* and H3K9 KMT 6KO samples was calculated relative to the wt sample (indicated above the bars). Fold increase of MSR RNA in the MSR-dCas9-Activator (-dox) and MSR-dCas9-Activator (+dox) samples was calculated relative to the MSR-dCas9-Control (−dox) sample (indicated above the bars). The dashed line indicates the average normalised counts for *Hprt* expression across all three datasets. The red segment indicates the threshold range of MSR transcript deregulation derived from proportional conversion of RT-qPCR values shown in (**b**) and is equivalent to 13,000–37,000 normalised RNA-seq counts.

---

with mitotic defects and genome instability. Indeed, DAPI confocal imaging of the MSR-dCas9-Activator cell population 24 h after doxycycline removal showed that the majority of cells display hallmarks of apoptosis and genomic instability (cell enlargement, apoptotic bodies, multinucleated cells and micronuclei) (Fig. 7d). Taken together, these data indicate that a 6 h induction of MSR-dCas9-Activator irreversibly compromises cell viability that is reflected by mitotic defects, high-levels of apoptosis and genomic instability.

## Discussion

### Enforced silencing of MSR transcription

MSR transcription is needed for the developmental progression of the mouse pre-implantation embryo and has been associated with accumulated MSR transcripts to direct establishment and organisation of pericentric heterochromatin[15,16]. Our results support a role for MSR transcripts in the structural organisation of pericentric heterochromatin. When MSR RNA is reduced, either by targeting TALE-Onconase-EGFP to MSR sequences in mESC or by silencing MSR transcription with the MSR-dCas9-Repressor in MEF cells, heterochromatic regions aggregate. This could primarily be caused by increased recruitment of components of the silencing machinery and protein-based coalescence of chromatin domains that contain these silencing factors. In addition, our data with MSR targeted TALE-Onconase-EGFP suggest that MSR RNA may itself also have a function in the structural organisation of heterochromatin. An RNA-based mechanism for a heterochromatic scaffold has been proposed[8,13], and it is possible, although currently not resolved, that reduced MSR RNA levels will allow for a more clustered focal arrangement of heterochromatin domains.

A recent report showed that a 50% reduction of MSR RNA in mESC resulted in an increase in the number of DAPI-dense regions[9]. This appears to be in contrast to our work with TALE-Onconase-EGFP where we observe less but enlarged DAPI-dense regions. A possible reason for these divergent data may be due to differences in the cultivating conditions of mESC, which can influence their pluripotent state. In our study, we used serum-free media to maintain a high level of mESC pluripotency. Consistent with other reports, the number of DAPI-dense regions are lower at high mESC pluripotency, but increase when mESC start to lose pluripotency[40–42].

### A threshold limit for MSR deregulation

In contrast to enforced silencing of MSR RNA, focal organisation of heterochromatin is disrupted by massive overexpression of MSR RNA. We used our experimental approach to define a threshold limit of >300-fold deregulation of MSR transcript levels. This threshold limit can explain why in the uninduced MSR-dCas9-Activator cells, a 40-fold increase in MSR transcript levels does not disperse DAPI-dense regions. It also relates that physiologically elevated MSR transcript levels (50–120-fold) during early mouse embryogenesis are below the

threshold limit and do not break but rather assist heterochromatin formation[16]. Aberrantly high levels for MSR transcripts are found in several mouse tumour models[17,34] and satellite RNA is frequently overexpressed in many forms of human cancer[18,19]. We note that the threshold limit for MSR deregulation has been defined in MEF cells and may fluctuate in other cell types, particularly if they differ in the amount or composition of heterochromatin components. Our definition of a threshold limit for MSR deregulation and its implication for inducing mitotic defects and genomic instability helps to provide insight for pathological states of overexpressed MSR transcripts. Persistent overexpression of MSR RNA even after the levels of MSR-dCas9-Activator has declined resulted in impaired chromosome segregation, apoptosis and cell death. Highly elevated MSR transcripts in *Dicer*-deficient mESC have been suggested to be causative for chromosome segregation defects and reduced viability[43]. Our data are consistent with a role of deregulated MSR transcription in driving genome instability and also indicate that a massive over-production of MSR transcripts, significantly above the threshold limit, are lethal and cannot be reverted. We like to suggest that controlled MSR RNA output below the threshold limit can form an RNA-nucleosome scaffold[11,13] or an RNA-chromatin scaffold[8] to stabilise heterochromatin structure, while a massive overshoot of MSR RNA disrupts the structural organisation of heterochromatin.

### A transcriptionally hyperactive chromatin state drives heterochromatin disruption

The MSR-dCas9-Activator mediated heterochromatin disruption is similar to perturbed heterochromatin organisation following targeting of transcriptional activators to the nuclear lamina[44] or to core heterochromatin[10]. In all these cases, heterochromatin disruption is associated with an increase in histone H3 acetylation. Further, a global increase in histone acetylation through the use of HDAC inhibitors has been shown to redistribute DAPI-dense regions to the nuclear periphery and to impair heterochromatin organisation[45]. If histone acetylation is inhibited during induction of MSR-dCas9-Activator, dispersion of DAPI-dense regions is prevented, but MSR overexpression is also impeded. By inhibiting both RNA transcription and histone acetylation during the induction of MSR-dCas9-Activator, an additive effect on preventing heterochromatin disruption is observed. While we are not excluding a role for MSR RNA, these data suggest that a transcriptionally hyperactive chromatin state is the major driver for the disruption of heterochromatin. As with an MSR RNA threshold limit, there may also be a threshold limit for histone acetylation and associated transcriptional activators above which heterochromatin organisation becomes disrupted. This is in agreement with past research showing that the insertion of strong enhancers into heterochromatic regions[46] or enforcing transcriptional activity within heterochromatin[10] can override heterochromatin silencing. To determine whether and/or to what degree MSR RNA output per se is

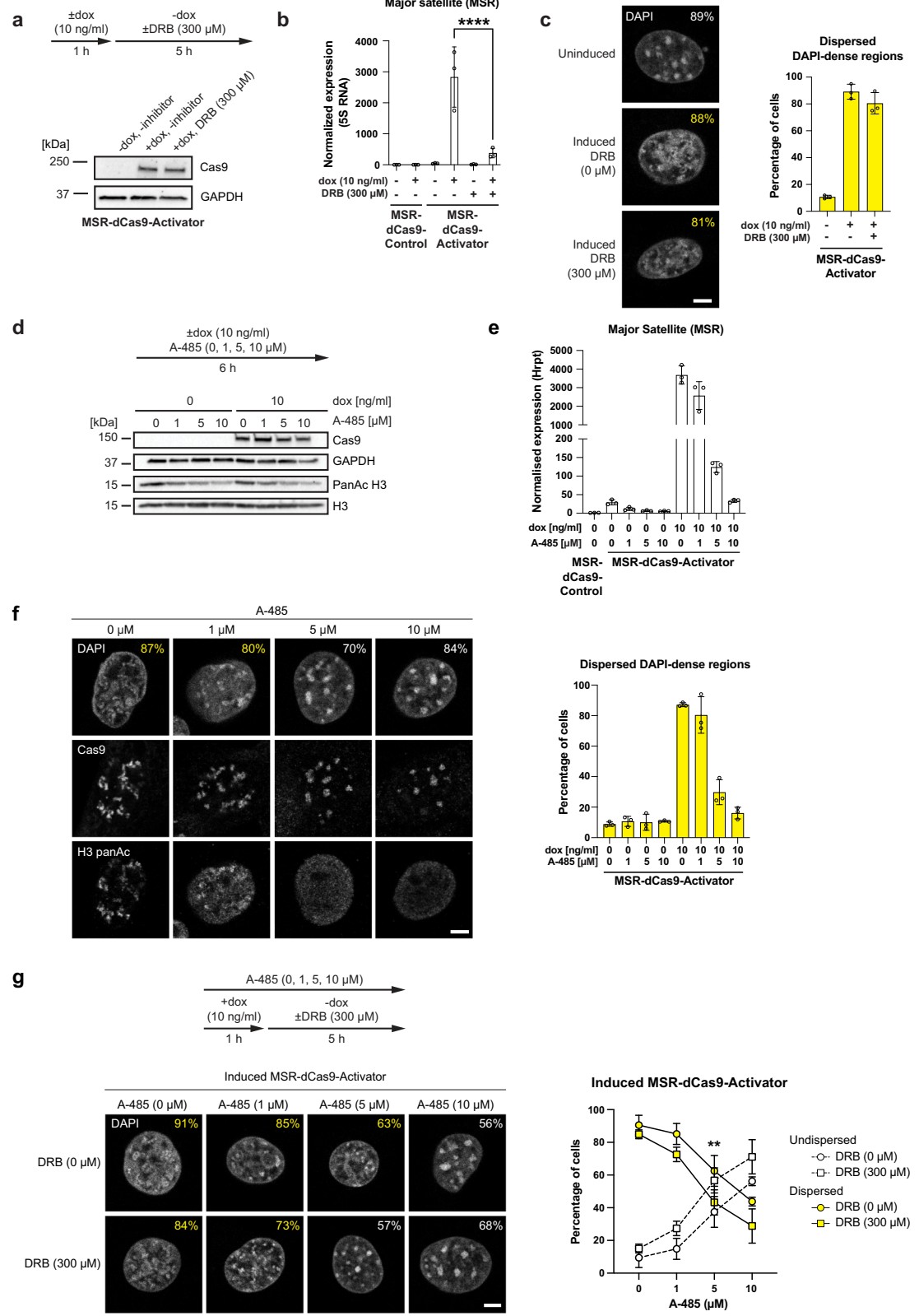

influencing the structural organisation of heterochromatin, a transcriptionally hyperactive chromatin state would need to be uncoupled from the process of transcription. This could possibly be approached through the use of a single-unit RNA polymerase, such as T7 RNA polymerase, which does not require transcriptional co-factors or an activated chromatin state (but it requires chromatin access). It would be interesting to explore whether an inducible MSR-dCas9-T7 RNA polymerase could be targeted to mouse pericentric heterochromatin and generate MSR transcripts.

## MSR DNA repeat arrays as a solid heterochromatin scaffold

A great amount of research has focused on how pericentric heterochromatin is organised[2,6]. One model suggests that liquid-liquid phase separation drives heterochromatin formation[9,47,48]. Other biophysical

**Fig. 5 | RNA transcription and hyperacetylated chromatin drive heterochromatin disruption. a** Western blot analysis for expression of MSR-dCas9-Activator in MEF cells (see flow diagram for induction and inhibition conditions). GAPDH expression is shown as a loading control. **b** RT-qPCR analysis for MSR transcripts in MEF cells treated as in (**a**). Values were normalised to 5S RNA and are relative to MSR-dCas9-Control (−dox, −DRB) (mean ± SD) (****, $p \leq 0.0001$, one-way ANOVA, Tukey's test). $n = 3$ independent experiments. **c** Confocal imaging of DAPI-dense regions in MSR-dCas9-Activator MEF cells that were treated as in (**a**). Scale bar is 5 μm. The percentages reflect the fraction of cells with either undispersed (white) or dispersed (yellow) DAPI-dense regions. For each condition $n \geq 200$ cells were analysed from $n = 3$ independent experiments. Quantification of the imaging data is shown on the right (mean ± SD). **d** Western blot analysis for expression of MSR-dCas9-Activator and pan-acetylated H3 in MEF cells (see flow diagram for induction and inhibition conditions). GAPDH and H3 expression are shown as loading controls. **e** RT-qPCR analysis for MSR transcripts in MEF cells treated as in (**d**). Values were normalised to *Hprt* and are relative to the MSR-dCas9-Control (−dox,-A-485) (mean ± SD). $n = 3$ independent experiments. **f** Immunofluorescence of MSR-dCas9-Activator MEF cells treated as in (**d**). Cells were immunolabelled for Cas9 and pan-acetylated H3 and counterstained with DAPI. Scale bar is 5 μm. The percentages reflect the fraction of cells with either dispersed (yellow) or undispersed (white) DAPI-dense regions. For each condition $n \geq 135$ cells were analysed from three independent experiments. Quantification is shown on the right (mean ± SD). **g** Confocal imaging of DAPI-dense regions after combined inhibition of RNAPII and HAT enzymes in induced MSR-dCas9-Activator MEF cells (see flow diagram). Scale bar is 5 μm. The percentages reflect the fraction of cells with either dispersed (yellow) or undispersed (white) DAPI-dense regions. The total number of cells analysed per condition are as follows: (−DRB,−A-485) $n = 159$, (−DRB,1 μM A-485) $n = 197$, (−DRB,5 μM A-485) $n = 172$, (−DRB,10 μM A-485) $n = 205$, (+DRB,−A-485) $n = 157$, (+DRB,1 μM A-485) $n = 186$, (−DRB,5 μM A-485) $n = 172$, (−DRB,10 μM A-485) $n = 220$. $n = 3$ independent experiments. Quantification of the imaging data is displayed on the right (mean ± SD) (**, $p \leq 0.0044$, two-way ANOVA, Šídák's test).

models propose that (hetero)chromatin is a polymer[10,49] and can be considered as an immobile solid[50] where liquid (hetero)chromatin factors coalesce and form phase-separated compartments[10,49]. While reconstituted nucleosomes can undergo liquid-liquid phase separation that is partially antagonised by histone acetylation[51], histone acetylation disperses but does not liquify chromatin in living cells[49].

If RNA-protein condensates and HP1 are drivers for heterochromatin organisation, then loss of HP1 should result in disordered DAPI-dense regions. Indeed, mESC triple-deficient for HP1α, HP1β, and HP1γ show dispersion of DAPI-dense foci but also have significantly reduced H3K9me3 signals[52]. This is consistent with a previous study in MEF cells ablating all six H3K9 methyltransferase enzymes (KMT). In these 6KO H3K9 KMT mutant MEF cells, DAPI-dense regions were highly disrupted and had lost all H3K9 methylation and HP1 localisation[35]. In addition, MSR transcripts (and transcripts of other repeat classes) were massively derepressed[35]. In contrast, Suv39h double-null MEF cells also lack pericentric H3K9me3 and HP1α localisation but up-regulate MSR transcripts by only less than 5-fold. Intriguingly, *Suv39h* double-null MEF cells preserve the focal organisation of DAPI-dense regions, although HP1α localisation is nuclear diffuse[35,36]. Targeting of a dCas9-VPR to heterochromatin in *Suv39h* double-null MEF cells has been shown to disperse DAPI-dense regions[10]. Thus, the focal organisation of pericentric heterochromatin may involve other mechanisms independent of HP1α which, however, also appear to be influenced by an MSR RNA overshoot and an activated chromatin state of MSR repeats.

In the current study, dispersed DAPI-dense regions following the induction of the MSR-dCas9-Activator still maintained their 'heterochromatic' signature, i.e. the presence of H3K9me3 and localisation of HP1 proteins. Regardless of how high the levels of MSR transcripts are modulated, HP1 remains associated with the dispersed DAPI-dense (i.e. A/T-rich MSR repeat DNA) regions. HP1 proteins have been shown to dimerise[53] and stabilise a heterochromatic nucleosome structure[54]. HP1 also binds to RNA[55] and HP1α association with dispersed DAPI-dense regions is lost by RNase A incubation of permeabilised MSR-dCas9-Activator MEF cells. In a model, where a more compact heterochromatin scaffold is largely formed by multivalent HP1-HP1 contacts, an overshoot of MSR RNA would saturate HP1 RNA binding and disfavour inter-nucleosomal protein-protein interactions. This would be consistent with RNA to be a buffer in modulating phase separation of RNA-protein condensates and where high RNA concentrations counteract the transition to more solid-like assemblies[56,57].

Mouse pericentric heterochromatin in *M. musculus* is characterised by large arrays of reiterated copies of the MSR consensus unit (234 bp)[3]. These A/T-rich MSR repeat arrays can incorporate >10,000 copies of major satellite sequences at the pericentric regions of chromosomes[26,58] and are visualised as the classic DAPI-dense regions. This focal organisation of pericentric heterochromatin in *M.*

*musculus* is likely to be further enhanced by the biophysical properties of the A/T-rich MSR DNA and the reiterative nature of MSR repeat units which has been proposed to adopt a non-B-form DNA configuration[59] and to allow for RNA:DNA hybrid formation[13]. The arrangement and DNA sequence of major satellite repeats is not conserved in closely related mouse (e.g. *M. spretus*) or other rodent species and also not in human heterochromatin[60]. Strikingly, MEF cells from *M. spretus* do not have focal heterochromatic regions[60]. Focal pericentric heterochromatin organisation follows MSR repeat abundance as observed in *M. musculus* x *spretus* hybrid oocytes.[61] Our data support a model for mouse pericentric heterochromatin organisation, where the underlying A/T-rich MSR repeat DNA and associated MSR RNA form a polymer scaffold that dictates the spatial distribution of HP1 and the dynamic folding of heterochromatic foci.

In summary, our study highlights the importance of restricted MSR RNA output and provides insights for the structural organisation of pericentric heterochromatin and MSR RNA modulation of genome stability.

## Methods

### Generation of inducible MSR-dCas9-effector components
To generate inducible dCas9-effector components the following plasmids were obtained from Addgene: pB-TRE-dCas9-VPR (63800), pB-CAGGS-dCas9-KRAB-MeCP2 (110824), pB-CAGGS-dCas9 (118023) and pB-U6insert (104536). The plasmids pB-CAGGS-dCas9-KRAB-MeCP2 and pB-CAGGS-dCas9 were digested with AseI and PmeI. This plasmid fragment was then inserted into the AseI-PmeI digested pB-TRE-dCas9-VPR to create the inducible version of dCas9-KRAB-MeCP2 and dCas9; pB-TRE-dCas9-KM and pB-TRE-dCas9. The gRNA sequence to target MSRs, GAAATGTCCACTGTAGGACG, was PCR amplified with the scaffold sequence and cloned into the AgeI and EcoRI restriction sites of the plasmid pB-U6insert. This newly generated plasmid containing the gRNA and scaffold was then PCR amplified to obtain a DNA fragment containing the U6 promoter and the gRNA and scaffold. This fragment was digested with SpeI and cloned into the SpeI site of pB-TRE-dCas9-KM and pB-TRE-dCas9. The final plasmids are: pB-TRE-gMSR3-dCas9, pB-TRE-gMSR3-dCas9-KM and pB-TRE-gMSR3-dCas9-VPR where the gRNA is constitutively expressed within the same plasmid backbone as the inducible dCas9-effectors. See Supplementary Table 1 for oligonucleotide and primer sequences.

### Generation of inducible MSR-dCas9-effector MEF cells and cell culture
Inducible MSR-dCas9-effector MEF cells were created with the PiggyBac transposon system (SBI) following the manufacturer's protocol. In brief, $7.5 \times 10^4$ cells were plated per well of a 6-well plate. The next day, the PiggyBac plasmid was transfected together with either pB-TRE-gMSR3-dCas9, pB-TRE-gMSR3-dCas9-KM or pB-TRE-gMSR3-dCas9-

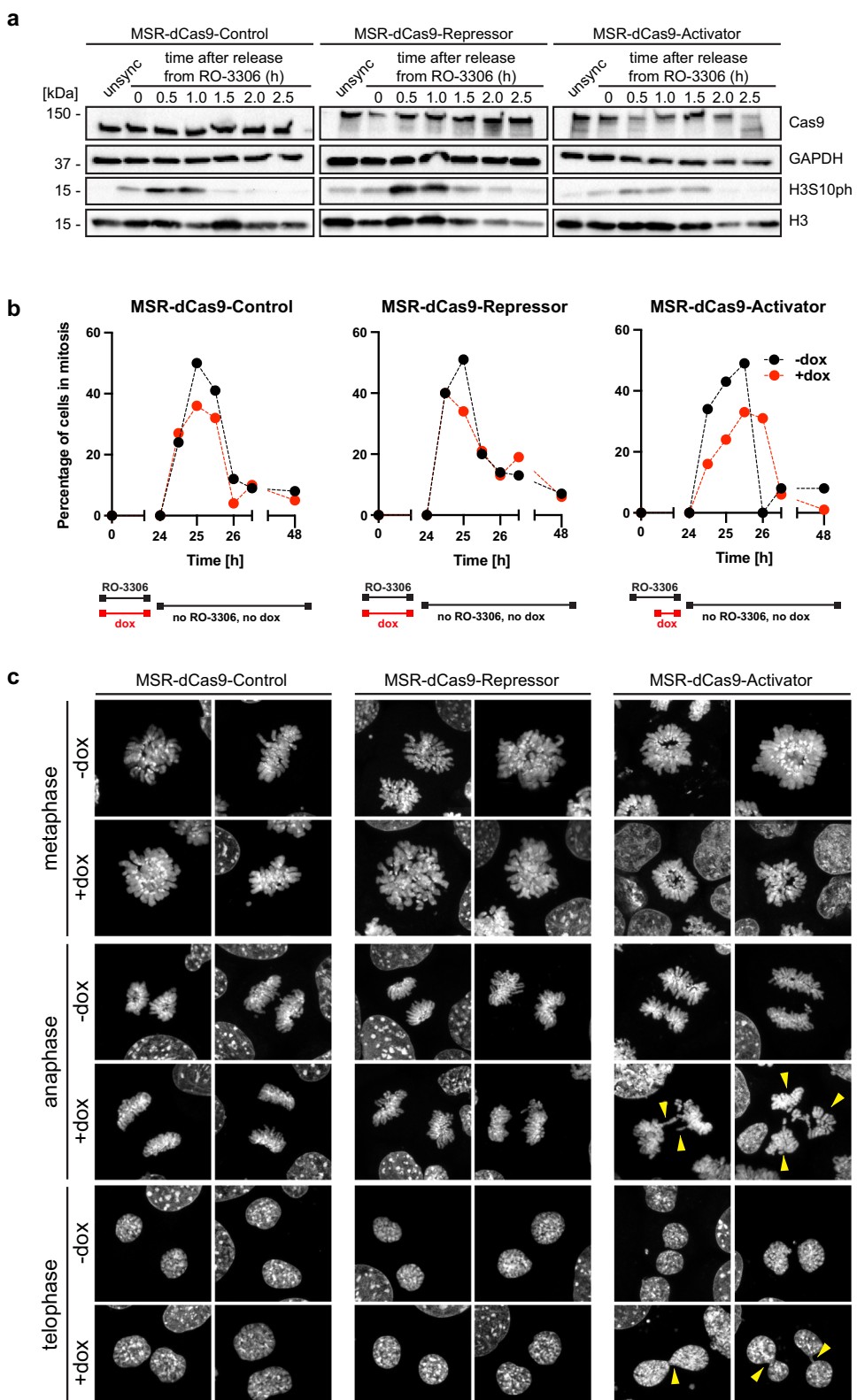

VPR using Lipofectamine 3000 to generate MSR-dCas9-Control, MSR-dCas9-Repressor and MSR-dCas9-Activator MEF cell lines. Of, 72 h post-transfection, selection was carried out in complete medium supplemented with 400 µg/ml hygromycin until confluency was reached. MSR-dCas9-effector cells were maintained as mixed cell cultures. MSR-dCas9-effector expression was confirmed by western blot and immunofluorescence microscopy using the Cas9 antibody. Cells were maintained in high glucose DMEM (Sigma-Aldrich) containing 10% tet-free FBS (Cytiva), 2 mM L-glutamine (Sigma-Aldrich), 0.1 mM β-mercaptoethanol, 1x non-essential amino acids (Sigma-Aldrich), 1 mM sodium pyruvate (Sigma-Aldrich), and 400 µg/ml hygromycin. For induction studies, MEF cells were treated with 10 or 100 ng/ml doxycycline in complete medium without hygromycin for 6 or 24 h as indicated in the text.

**Fig. 6 | Delayed mitotic progression and chromosome mis-segregation in MEF cells expressing MSR-dCas9-Activator. a** Western blot analysis for detection of induced MSR-dCas9-Control, MSR-dCas9-Repressor and MSR-dCas9-Activator in unsynchronised and in RO-3306 synchronised MEF cells. Following release from RO-3306 inhibition, cell samples were collected every 30 min for an interval of up to 2.5 h. Mitotic progression was verified by probing for histone H3S10 phosphorylation (H3S10ph). GAPDH and H3 are shown as loading controls. **b** Profiles for mitotic progression of RO-3306 synchronised MSR-dCas9-Control, MSR-dCas9-Repressor and MSR-dCas9-Activator MEF cells without doxycycline (-dox, black circles) or with doxycycline (+dox, red circles) induction. Following release from RO-3306 inhibition, cell samples were collected every 30 min for an interval of up to 2.5 h. Percentages of mitotic cells were quantified by immunofluorescence for H3S10 phosphorylation. For each sample $n \geq 60$ cells were analysed. **c** Confocal imaging of chromosomes in metaphase, anaphase or telophase stages from MSR-dCas9-Control (left panel), MSR-dCas9-Repressor (middle panel) and MSR-dCas9-Activator (right panel) MEF cells without doxycycline (-dox) or with doxycycline (+dox) induction. Chromosomes were counterstained with DAPI. For each mitotic stage, two representative images are shown. Scale bar is 10 μm. Yellow arrows indicate lagging chromosomes, chromosome bridges, multiple mitotic spindles and non-segregated chromosome sets in MEF cells expressing MSR-dCas9-Activator. Quantification of the mitotic defects is shown in Supplementary Fig. S7.

## Flow cytometry analysis for cell viability and cell cycle

Cells treated with or without doxycycline were harvested by trypsinization and then stained with a Pacific Blue-conjugated anti-Annexin V antibody (Invitrogen, A35122, 1:500) and SYTOX Orange (Invitrogen, S11368, 10 nM) according to manufacturer's protocol. Cell suspensions were then filtered through a 35 μm mesh. $1.0 \times 10^5$ cells were analysed using the BD FACSymphony A5 analyzer (BD Bioscience). Data were analysed with the use of FlowJo and GraphPad Prism software. Gating strategy is shown in Supplementary Fig. 9a.

For cell cycle analysis, cells were plated and treated with or without doxycycline. After induction, cells were harvested, washed with PBS and pelleted. The cell pellet was fixed with 1% paraformaldehyde in PBS for 10 min at room temperature and then neutralized with glycine (final concentration of 125 mM). Cells were washed with FACS buffer (PBS, 0.5% BSA, 5 mM EDTA). Cells were pelleted and resuspended in permeabilization buffer (00-8333-56, Invitrogen) containing APC-eFluor780 conjugated anti-Ki-67 antibody (47-5698-82, Invitrogen, 1:200) and DAPI (1:1000) and incubated for 30 min on ice. Cells were washed with permeabilization buffer and resuspended in FACS buffer. Cell suspensions were then filtered through a 35 μm mesh prior to flow cytometry. $1.0 \times 10^5$ cells were analysed using the BD FACSymphony A5 analyzer (BD Bioscience). Data were analysed with the use of FlowJo. Gating strategy shown in Supplementary Fig. 9b. Flow cytometry experiments were performed in triplicate.

## RNA polymerase II and histone acetyltransferase (HAT) inhibitor treatments

$2.0 \times 10^5$ cells were plated per well of a 6-well dish. To inhibit RNA polymerase II, cells were first induced with doxycycline for 1 hr. After induction, the cells were washed with PBS and replaced with fresh media containing 300 μM 5,6-Dichlorbenzimidazol 1-β-D-Ribofuranosid (DRB) (Sigma-Aldrich, D1916) for 5 h. To inhibit HAT activity, cells were induced with doxycycline in the presence of 1, 5, or 10 μM of A-485 (Sigma-Aldrich, SML2192). To inhibit both RNA polymerase II activity and HAT activity, cells were first induced with doxycycline for 1 h in the presence of A-485 (0, 1, 5, or 10 μM). Cells were then washed with PBS and replaced with fresh media containing A-485 (0, 1, 5, or 10 μM) with or without 300 μM DRB for 5 h.

## Cell cycle synchronisation with RO-3306

Of, $1.0 \times 10^5$ cells were plated per well of a 6-well dish. The next day, medium was exchanged to synchronise cells at the G2/M phase of the cell cycle and induced for the expression of the MSR-dCas9-effector components. Medium containing 1 mM RO-3306 (Sigma-Aldrich, SML0569) for uninduced cells, medium containing 1 mM RO-3306 and 10 ng/ml of doxycycline for MSR-dCas9-Control MEF cells, or medium containing 1 mM RO-3306 and 100 ng/ml of doxycycline for MSR-dCas9-Repressor MEF cells was exchanged and incubated for 24 h. MSR-dCas9-Activator MEF cells were incubated with medium containing 1 mM RO-3306 for 18 h and then with medium containing both 1 mM RO-3306 and 10 ng/ml of doxycycline for the final 6 h. Unsynchronised and RO-3306 synchronised MSR-dCas9-effector MEF cells were processed for flow cytometry cell viability assay. To analyse cell viability post-RO-3306 synchronisation, samples were collected 24 h after removal of RO-3306 and processed for immunofluorescence microscopy and flow cytometry cell viability assay. To analyse cell viability under continued RO-3306 synchronisation, RO-3306-synchronised induced and uninduced cells were transferred to medium containing RO-3306 and without doxycycline for 24 h (RO-3306 blocked). Samples were then processed for flow cytometry cell viability assay and immunofluorescence microscopy.

## Preparation of mitotic chromatin stages

To monitor cells as they progress through mitosis from a G2/M-synchronisation, RO-3306-synchronised induced or uninduced cells were transferred to medium without RO-3306 or doxycycline and harvested every 30 min for the next 2.5 h. Cell samples were then processed for western blotting or immunofluorescence microscopy as described below.

## Cellular extracts and western blot analysis

Of, $5.0 \times 10^4$ cells were plated per well of a 6-well plate. The three MSR-dCas9-effector cell lines were induced in such a way that all induction time lengths would be complete at the same time 24 h later. Cells were harvested with trypsin, pelleted and washed with PBS. Cell pellets were lysed in 2X Laemmli buffer. For the mitotic analysis, cells were washed with PBS and then lysed in RIPA buffer. Samples were then sonicated (30 s ON, 30 s OFF, 7 cycles) (Bioruptor, Diagenode) and centrifuged at maximum speed at 4 °C for 10 min to recover the supernatant. SDS-PAGE analysis was conducted according to standard procedures, with antibodies against Cas9 (Cell Signalling, 14697, 1:1000), GAPDH (Santa Cruz, sc-32233, 1:2000), H3S10ph (Millipore, 06-570, 1:500), H3 (Abcam, ab1791, 1:50000), H3K4me3 (Abcam, ab8580, 1:5000), H3K9me3 (Abcam, ab8898, 1:20000), H3K27me3 (Diagenode, C15410195, 1:2000), H3K36me3 (Abcam, ab9050, 1:2000), and H3 pan-acetyl (Abcam, ab47915, 1:5000). Blots were imaged using a ChemiDoc system (Bio-Rad) and analysed using Image Lab programme (Bio-Rad). Western blots were performed in triplicate.

## Immunofluorescence microscopy

Cells were fixed and permeabilised in cold methanol at −20 °C for 20 min. After rehydration and washing with PBS, samples were incubated with antibodies diluted in PBS containing 5% donkey serum (Jackson ImmunoResearch, 017-000-121). H3K4me3 (Abcam, ab8580, 1:2000), H3K9me3 (Abcam, ab8898, 1:2000), H3K27me3 (Diagenode, C15410195, 1:2000), H3K36me3 (Abcam, ab9050, 1:2000), H3 pan-acetyl (Abcam, ab47915, 1:2000), Cas9 (Cell Signalling, 14697, 1:800), HP1α (Abcam, ab203432, 1:500), HP1β (Cell Signalling, 8676, 1:800), HP1γ (Cell Signalling, 26193, 1:100), RNA Pol II Ser5phos (Cell Signalling, 13523, 1:2000), CREST (Immunovision, HCT-0100, 1:500), and HMGA1 (Abcam, ab129153, 1:2000). After antibody labelling, samples were mounted with VECTASHIELD (Vector Laboratories, H-1200-10) containing DAPI and imaged at 63x magnification with an LSM900-Airyscan2 confocal microscope (Zeiss) using the Airyscan super-resolution (SR) mode and ZEN blue software (Zeiss).

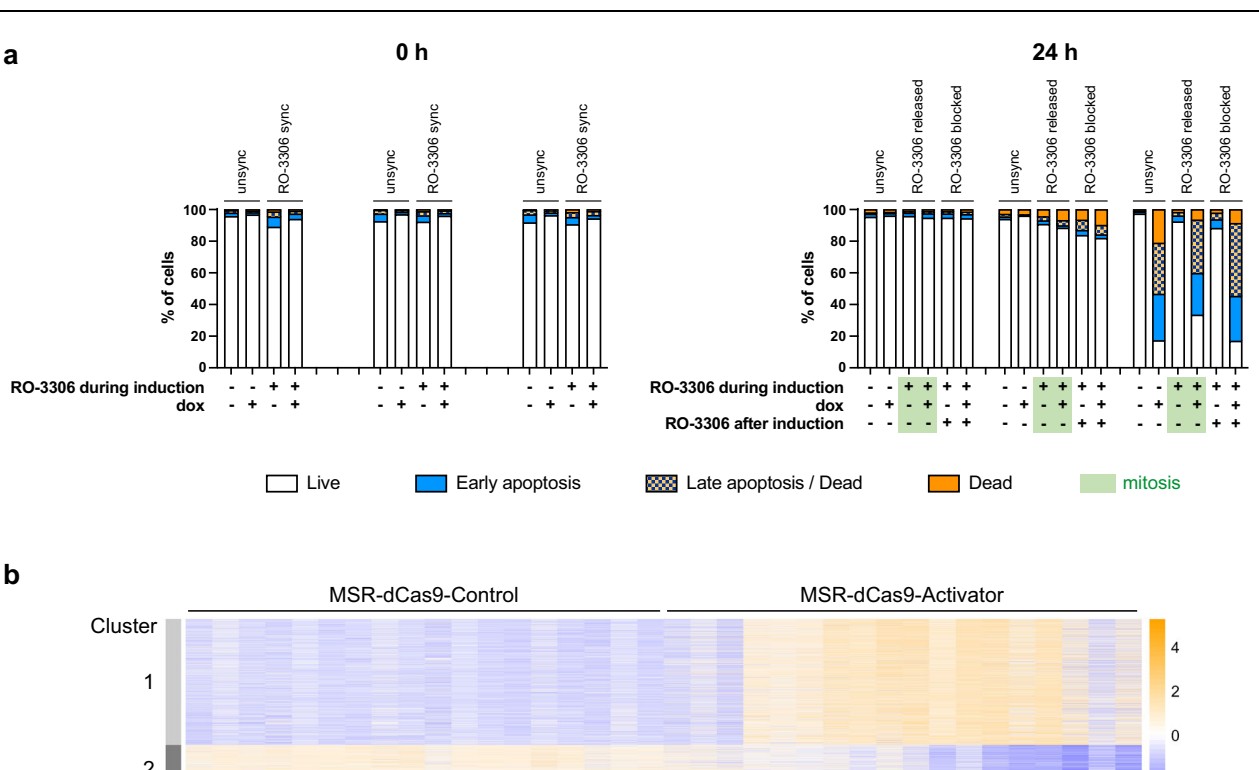

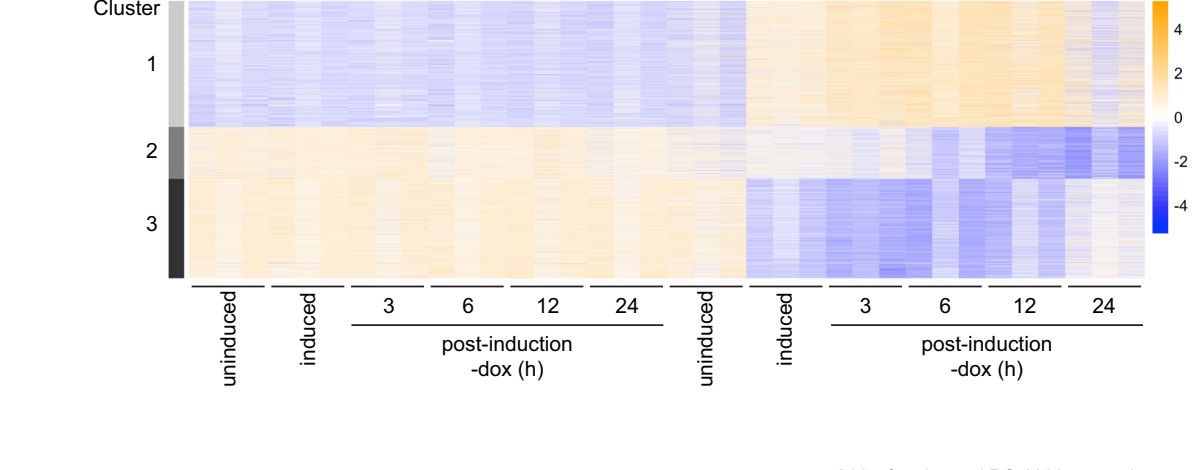

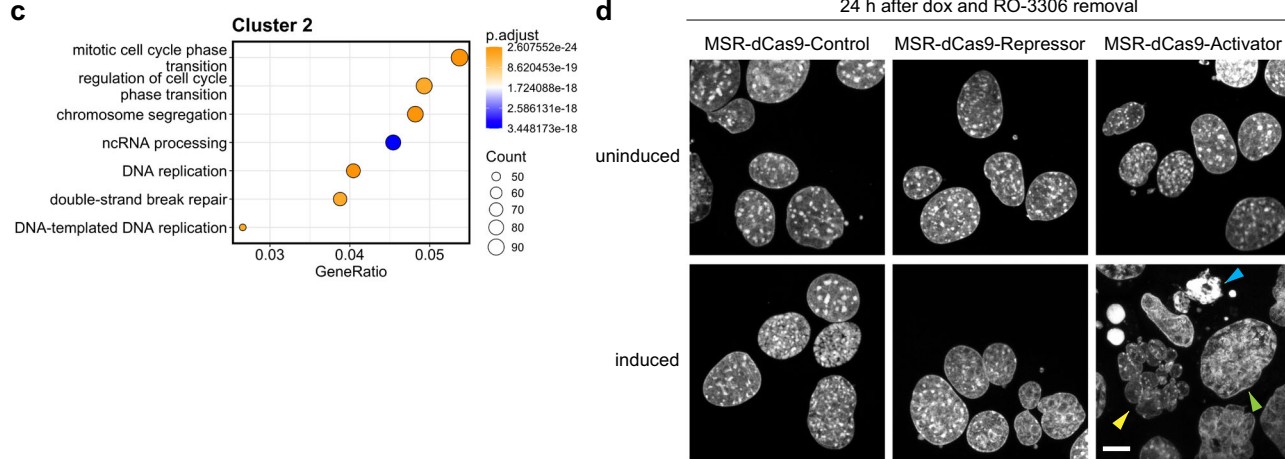

**Fig. 7 | Induced expression of MSR-dCas9-Activator irreversibly compromises cell viability. a** Bar graphs depicting percentages of live (white), apoptotic (blue) or dead (orange) cells in MSR-dCas9-Control, MSR-dCas9-Repressor and MSR-dCas9-Activator MEF cell populations either 0 h (left panel) or 24 h (right panel) after doxycycline removal. Cell viability was analysed for unsynchronised or RO-3306 synchronised cells and also when cells were RO-3306 synchronised and released or RO-3306 synchronised and blocked. Highlighted by light green shading (for the 24 h time point) are cell culture conditions where cell samples were released into mitosis. **b** Heatmap of differentially expressed genes between MSR-dCas9-Control and MSR-dCas9-Activator MEF cells. Uninduced, induced and post-induction (3, 6, 12 and 24 h after dox removal) conditions were compared. Differentially expressed

genes were filtered with a basemean >100 reads, >2-fold expression change and a p-adjusted value < 0.05. k-means clustering revealed that differentially expressed genes between MSR-dCas9-Control and MSR-dCas9-Activator during the recovery phases group into three clusters. *n* = 3 independent experiments. **c** Gene ontology (GO) analysis of **c**luster 2. Dot plots display the top 7 GO terms. *n* = 3 independent experiments. GO analysis of clusters 1 and 3 are shown in Supplementary Fig. 8c. **d** Confocal imaging (DAPI counterstaining) of induced MSR-dCas9-Control, MSR-dCas9-Repressor and MSR-dCas9-Activator MEF cell populations 24 h after doxycycline (dox) removal. Scale bar is 10 μm. Indicated with arrows are micronuclei and multi-nucleated cells (yellow), apoptotic bodies (blue) and enlarged nuclei (green) from cells induced to express MSR-dCas9-Activator.

To determine the percentages of undispersed vs dispersed DAPI-dense regions for the doxycycline titration and doxycycline induction time course experiments, single-blinded counts from three independent experiments were performed. MEF cells were considered to have dispersed DAPI-dense regions if they displayed non-focal DAPI patterns.

For the analysis of mitotic chromatin stages, cells were immobilized on slides by Cytospin (Thermo Scientific), washed with PBS, fixed at room temperature with 4% PFA for 15 min and permeabilised with 0.5% Triton X-100 for 5 min. Samples were then stained with an antibody against H3S10ph (Millipore, 06-570, 1:500) and mounted with VECTASHIELD (Vector Laboratories, H-1200-10) containing DAPI. Mitotic chromatin stages were imaged at ×63 magnification with an LSM780 confocal microscope (Zeiss). Maximum intensity projections of the mitotic stages were made with the ZEN Black software (Zeiss). The percentage of mitotic cells was calculated as a fraction of H3S20ph-positive cells to all cells.

### Indirect immunofluorescence and DNA FISH

Cells were grown on glass coverslips, washed with PBS, fixed at room temperature with 2% paraformaldehyde (PFA) in PBS for 10 min and permeabilised with 0.5% Triton X-100 for 5 min. Immunofluorescence was performed as described above, with antibodies against Cas9 (Cell Signalling, 14697, 1:800). Samples were post-fixed for 10 min with 4% PFA in PBS prior to DNA FISH detection. Samples were dehydrated with increasing concentrations of ethanol (70%, 90%, 100%) and left to air dry. LNA probes against MSRs (see Supplementary Table 1) were diluted to a final concentration of 3 nM for each probe in hybridisation solution (2X SSC, 50 mM sodium phosphate pH 7.0, 60% formamide). The probe solution was added to the samples and covered with a coverslip and denatured at 80 °C for 3 min. Probe hybridisation occurred at 37 °C for 1 h. Slides were washed in 2X SSC in 60% formamide at 37 °C. Then subsequently washed at room temperature with 2X SSC. Blocking buffer (4X SSC, 0.1% Tween 20, 5% BSA) was added to the slides for 15 min. FISH probes were detected with Streptavidin-A555 (Invitrogen, 521381, 1:1000) diluted in blocking buffer. Slides were washed in 2X SSC and subsequently mounted with VECTASHIELD (Vector Laboratories, H-1200-10) containing DAPI.

### RNA FISH analysis for strand-specific MSR repeat transcripts

RNA FISH[62] was performed with the following modifications. Cells were grown on glass coverslips and were uninduced or induced with doxycycline, washed with cold PBS, and subsequently washed with cold CSK buffer (100 mM NaCl, 3 mM MgCl$_2$, 300 mM Sucrose, 10 mM PIPES pH 6.8) containing 10 mM RVC (Ribonucleoside-Vanadyl Complex, NEB, S1402) for 30 s. Cells were then permeabilised with CSK buffer containing 10 mM RVC and 0.5% Triton X-100 for 5 min on ice. After washing with CSK containing 10 mM RVC, cells were fixed with 4% PFA in PBS for 10 min. After washing with PBS, samples were treated without RNase A or treated with 1 mg/ml RNase A (Invitrogen, EN0531) in PBS for 20 min at room temperature. After RNase treatment, samples were dehydrated in an increasing gradient of ethanol concentrations (70%, 80%, 90%, 100%) and left to air dry. LNA probe mixes detecting either the purine-rich transcript (AS probe) or pyrimidine-rich transcript (S probe) (see Supplementary Table 1) were diluted in 100% formamide and denatured at 80 °C for 10 min and then placed on ice for 10 min. The probe mixes were then added to the samples and hybridised at 37 °C for 3 h. Samples were then washed with 2X SSC with 50% formamide at 37 °C, 2X SSC at 37 °C, 1X SSC at room temperature, and 4X SSC at room temperature. To detect the FISH probes, samples were incubated with Streptavidin-A555 (Invitrogen, 521381, 1:500) diluted in detection buffer (4X SSC, 1% BSA, 40 mM RVC). Samples were washed with 4X SSC, 4X SSC with 0.1% Triton X-100, and

subsequently with 4X SSC. Slides were mounted with VECTASHIELD (Vector Laboratories, H-1200-10) containing DAPI.

### Quantitative imaging and linescans

For the sphericity and volume measurements, optical z-stacks were obtained from samples prepared for DNA FISH detecting MSR using an LSM900-Airyscan2 confocal microscope (Zeiss) using the Airyscan super-resolution (SR) mode. Images were then processed with Imaris (9.7.0) microscopy analysis software. 3D masks were created from the MSR-FISH fluorescence channel and the volume and sphericity was then calculated. Data visualisation was performed in R (4.1.2) using ggplot2 (3.3.5)[63].

Linescans were performed using Fiji (2.9.0) on images of single optical slices obtained from an LSM900-Airyscan2 confocal microscope (Zeiss) using the Airyscan super-resolution (SR) mode. Using the line tool, a line of 20 μm in length and 20 pixels wide was drawn and pixel intensities in the various fluorescence channels were obtained and normalised. Linescan plots from these fluorescence intensity profiles were performed in R (4.1.2) using ggplot2 (3.3.5)[63]. Images were obtained from three independent experiments.

For the imaging analysis of undispersed vs dispersed DAPI-dense regions, MEF cells were counterstained with DAPI and single-optical slices in the z-axis were obtained. Cells were individually scored by visual inspection for either focal DAPI-dense regions (undispersed) or non-focal DAPI-dense regions (dispersed).

### RT-qPCR analysis for repeat transcripts

Cells were plated in 10 cm cell culture dish and treated with or without doxycycline. After harvesting the cells, total RNA was extracted with TRI Reagent (Sigma-Aldrich, 93289). The remaining DNA was digested with TURBO DNase (ThermoFisher Scientific, AM2238), followed by clean-up with RNA Clean & Concentrator (Zymo Research, R1013). cDNA was created using Maxima reverse transcriptase (ThermoFisher Scientific, EP0742) using random hexamers following the manufacturers protocol. qPCR mixes contained diluted cDNA with 2X SYBR Select Master Mix (ThermoFisher Scientific, 4472920) and 200 nM of target-specific forward and reverse primers (See Supplementary Table 1) in a total volume of 10 μl. qPCR was performed using a QuantStudio 6 Flex qPCR machine (Applied Biosystems). Cycle threshold (Ct) values were used to calculate normalised expression (ΔΔCt method).

### Purification of nucleic acids for RNA:DNA hybrid detection

Of, $2.5 \times 10^6$ induced or uninduced cells were harvested and washed in PBS. The pellet was resuspended in Lairds buffer (100 mM Tris pH8.5, 200 mM NaCl, 5 mM EDTA, 0.2% SDS). The samples were sonicated (45 s OFF, 15 s ON, 12 cycles) using a Bioruptor (Diagenode). After sonication, 0.2 mg/ml of proteinase K (ThermoFisher Scientific, AM2546) was added and incubated overnight at 55 °C. Total nucleic acids were purified with phenol:chloroform:isoamyl alcohol (Roth, A156.3), precipitated with 2 volumes of cold ethanol and 1/3 volume of 7.5 M ammonium acetate, and washed with 70% ethanol. The air-dried pellet and resuspended in nuclease-free water.

### RNA:DNA hybrid dot blot

In total, 7.5 μg of total nucleic acid was treated with 1.5 μl of RNase H (NEB, M0297) ( + RNase H) or with 1.5 μl of water (−RNase H) for 30 min at 37 °C. The treatment was repeated twice and purified using a Gel & PCR Clean up kit (Macherey & Nagel, 740609) and diluted to a final concentration of 50 ng/μl. In total, 100 ng of purified nucleic acids were spotted on Hybond N+ membrane (Amersham, RPN303B) and baked for 2 h at 80 °C. Detection was performed using standard immunoblot techniques. Membranes were rinsed in TBST (Tris buffered saline, 0.1% Tween 20) and blocking buffer (TBST, 5% milk powder) was added. The S9.6 antibody (Kerafast, Ab01137-23.0,

1:10000) and the secondary HRP-conjugated antibody were diluted in blocking buffer. All membrane washes were done with TBST. The membranes were developed with ECL (ThermoFisher Scientific, 32106) and imaged with a ChemiDoc system (Bio-Rad) and analysed using Image Lab programme (Bio-Rad). After ECL development, the membranes were washed with TBST, and stained with SYBR Gold nucleic acid stain (ThermoFisher Scientific, S11494, 1:200000) for 15 min and washed for 40 min. Images were obtained using the Axygen Gel Documentation system (Corning). Quantification of total nucleic acids was performed using Fiji (2.9.0).

### RNA:DNA immunoprecipitation (RDIP)
RDIP procedure was adapted from Sanz and Chédin[64] and described as follows. 10 µg of total nucleic acid was treated with 2 µl of RNase H (NEB, M0297) ( + RNase H) or with 2 µl of water (-RNase H) for 30 min at 37 °C. The treatment was repeated twice and the reaction was stopped with EDTA. 2 µg of the treated nucleic acids was diluted in TE buffer and 10% was kept as input. 10X DRIP binding buffer (100 mM sodium phosphate pH 7, 1.4 M NaCl, 0.5% Triton X-100) was added to the remaining 90% of the sample to make it 1X DRIP binding buffer. 2 µg of S9.6 antibody (Sigma-Aldrich, MABE1095) was and incubated overnight at 4 °C. 25 µl of protein G Dynabeads (ThermoFisher Scientific, 10003D) were washed in 1X DRIP binding buffer. The beads were then added to the samples and incubated for 2 h at 4 °C. Bead-antibody-hybrid complexes were then washed with 1X DRIP binding buffer at room temperature and eluted in DRIP elution buffer (50 mM Tris pH 8.0, 10 mM EDTA, 0.5% SDS). Proteinase K (ThermoFisher Scientific, AM2546) was added to a final concentration of 0.5 µg/µl and incubated for 45 min at 55 °C. Immunoprecipitated RNA:DNA hybrids were purified using a Gel & PCR Clean up kit (Macherey & Nagel, 740609).

### RNA sequencing
Total RNA was extracted with TRI Reagent (Sigma-Aldrich, 93289) and remaining DNA was digested with TURBO DNase (ThermoFisher Scientific, AM2238), followed by clean-up with RNA Clean & Concentrator (Zymo Research, R1013). Total RNA libraries were prepared using the TruSeq Stranded Total RNA Library Prep Gold (Illumina, 20020598). Paired-end, 150 bp reads were generated with Illumina HiSeq3000 sequencer with a depth of 50 million reads per sample. Three biological replicates were sequenced per cell line.

### RNA-seq analysis
Unprocessed reads were first quality and adapter trimmed using cutadapt[65] version 1.8.1 using a quality threshold of 20. Reads were then aligned to the GRCm38 genome from Ensembl using STAR[66] version 2.5.2b, with settings as recommended by TEtranscripts[67]. Both repeats and genes were then quantified using TEtranscripts version 2.0.3 and differentially expressed repeat elements determined using DESeq2 package version 1.34.0[68]. For analysis of repeat element expression, library size factors as computed from the gene expression data were used rather than recomputing these using the expression metrics of the repeat elements themselves. Data visualisation was performed in R version 4.1.2 using the ggplot2 version 3.3.5[63] and ggpubr version 0.4.0 packages.

To merge the datasets from this current study with the studies from Montavon et al. (2021) and Zhu et al. (2018), we used constant genes (genes that are not differentially expressed and have a basemean >10) to extract the size factors and then we applied the size factors to normalise the three datasets.

To extract differentially expressed genes for the RNA-seq data following MSR-dCas9-effector cells during and post-induction, DESeq2 was applied using the replicate, timepoint, and condition in the design formula. Genes that had a basemean >100, >2-fold change in expression, and an adjusted $p$ value < 0.05 were considered differentially expressed. K-means clustering and heatmap plotting was performed using pheatmap R package (1.0.12). Gene ontology analysis was performed using clusterProfiler (4.10.1)[69].

### Proportional conversion of RT-qPCR values to normalised RNA-seq counts
To calculate the threshold limit of MSR expression for the normalised RNA-seq counts, the lower and upper threshold limits obtained from RT-qPCR (Fig. 4b) was converted to an equivalent value of normalised RNA-seq counts. To determine the conversion factor (CF) the following formula was used:

$$CF = \frac{(\text{normalised counts RNA-seq})}{(\text{normalised RT-qPCR values})}$$

$$CF_{\text{avg}} = \frac{CF_{(-\text{dox, 0 ng/ml})} + CF_{(+\text{dox, 10 ng/ml})}}{2}$$

The average conversion factor was then applied to the RT-qPCR values to calculate the normalised RNA-seq counts for the 2 ng/ml and 3 ng/ml dox induction condition:

$$\text{normalised count} = (CF_{\text{avg}})(\text{normalised RT-qPCR value})$$

### Reporting summary
Further information on research design is available in the Nature Portfolio Reporting Summary linked to this article.

## Data availability
The RNA-seq datasets generated in this study have been deposited in the GEO repository under the accession code GSE287837. The data and any unique materials are available from the corresponding author upon reasonable request. All oligonucleotides used in this study are provided in Supplementary Table 1. Source data are provided with this paper.

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

## Acknowledgements

We thank Ulrike Bönisch, Stephanie Falk and the Deep Sequencing facility at the MPI-IE for processing samples; Roland Pohlmeyer and Sergiy Avilov from the Imaging facility at the MPI-IE for help with image acquisition and analysis; Sebastian Hobitz and Jörg Büscher from the Flow cytometry and DNA sequencing facility at the MPI-IE. We acknowledge the expert guidance of Maria-Elena Torres-Padilla in using TALE constructs to target MSR repeat sequences. We also thank the members of the Jenuwein laboratory for insightful discussions and for sharing reagents and protocols. Research in the laboratory of T.J. is supported by the Max Planck Society and by additional funds from the German Research Foundation (DFG) within the CRC992 consortium 'MEDEP'.

## Author contributions

Conceptualization, R.W.C. and T.J.; Methodology, R.W.C., K.M.S., B.K. and B.E.; Investigation, R.W.C. and K.M.S.; Formal Analysis, R.W.C., K.M.S. and G.E.; Writing – Original Draft, R.W.C. and T.J.; Supervision, T.J.

## Funding

## Competing interests

The authors declare no competing interests.
