## [Peer Review file · Nature Communications]

Forced expression of MSR repeat transcripts above a threshold limit breaks heterochromatin organisation

Corresponding Author: Dr Thomas Jenuwein

Version 0:

Reviewer comments:

Reviewer #1

(Remarks to the Author)

Here the authors set out to determine the consequences for heterochromatin formation of inducibly enforcing over- or under-expression of the major satellite repeat (MSR) RNA in mouse embryo fibroblasts (MEFs). For this purpose, they transduced MEFs using the PiggyBac transposon expression system and established cell lines in which the dCas9-VPR transcriptional activator and the dCas9-KM transcriptional repressor proteins could be inducibly expressed and targeted to MSR by a single gRNA that is constitutively expressed from the U6 promoter cassette integrated into the TRE-dCas9 plasmids. They showed that the Dox-induced dCas9 fusion proteins were targeted to pericentric heterochromatin, as monitored by DAPI and dCas9 staining, and that induction of the dCas9-VPR fusion protein led to a 4,000-fold increase in MSR-RNA and rapid cell death, whereas induction of the dCas9-KM fusion protein strongly reduced the level of MSR-RNA, with expression of the control dCas9 protein having no effect. A time course of the responses of cells to induction analyzed by staining with CREST anti-centromere antibodies showed that dCas9-VPR induced expression of MSR-RNA resulted in rapid dispersion of heterochromatin foci that still exhibited HP1 β , HP1 γ and HMGA1 co-localization, whereas induction of dCas9-KM caused delayed aggregation of heterochromatin. They also found that the levels of histone H3 pan-Ac and recruited RNA Pol II were elevated in dCas9-VPR activator domains. Analysis by immunofluorescence using strand-specific MSR probes showed that both the levels of both the forward and reverse MSR RNAs were elevated and remained localized to dCas9-VPR activator domains, and by probing a dot-blot with an antibody that detects RNA:DNA hybrids and carrying out RDIP they showed that MSR DNA-RNA hybrids were formed, validating the existence of RNA-DNA hybrids based on their disappearance upon prior treatment with RNase H. Finally, by using the RO-3306 CDK1 inhibitor to arrest cells in G2 followed by release into M phase, they showed that induced expression of dCas9-VPR delayed mitosis, as monitored by decreased pS10 H3 levels, and also led to chromosome mis-segregation, with a high frequency of lagging chromosomes. Prolonged induction of dCas9-VPR expression elicited MEF cell death and apoptosis, which were shown to be partially due to the observed mitotic defects, with other mechanisms also being responsible.

The use of inducible dCas9-activator and dCas9-repressor protein expression to regulate the expression levels of the major satellite repeat RNA in cells is an elegant approach to deciphering the function of MSR-RNA in pericentric heterochromatin organization. The experimental flow is logical and the conclusions are largely justified by the data, and support an important role for MSR transcripts in the structural organization of pericentric heterochromatin. Overall, these are interesting findings, although the mechanisms underlying the observed changes in heterochromatin structure have not been fully elucidated, and it would be interesting to know whether MSR RNA-scaffolded LLPS formation also occurs in the MEF cell system, as was reported for mESCs (ref. 11).

Points: 1. Figure 1: How stable are the dCas-fusion proteins, i.e. do they continue to accumulate following induction or do they reach a steady state level where of synthesis and turnover are balanced during these experiments? A schematic of the genomic organization of the MSR repeat and minor satellite region showing where potential TSS's map and where the gRNA target sites are localized would be helpful. In panel D, the level of MSR RNA was apparently 40-fold higher in the dCas9VPR cells than in the dCas9 control cells, even though according to the blots in panel A there does not appear to have been any dCas9-VPR protein in the uninduced cells. A constitutive 40-fold increase in MSR RNA level over that in control cells might have been expected to elicit some heterochromatin-dependent phenotypic changes. What is observed if the RNA expression patterns of the dCas9-VPR (-Dox cells) are compared to the dCas9 (-Dox) cells (panel E), i.e are there induced transcripts? Further discussion of these issues is warranted.

2. Figure 4: Given the huge increase in the levels of MSR RNA upon induction of the dCas9-VPR activator it seems unlikely that that much of it can form DNA:RNA hybrids - have the authors estimated the copy numbers per cell of the forward and reverse MSR RNAs upon induction compared to the number DNA copies of the MSR repeats? Does the RNase A sensitivity of the FISH signals in the dCas9-VPR activator panels mean that the forward and reverse strands RNA products are not forming dsRNA, or is dsRNA not denatured under the conditions used for RNA-FISH, and therefore not detectable. In this connection, can the authors test whether pretreatment with a dsRNase, such as RNase III, affects the total MSR RNA level?
3. Figure 5: Can the dCas-VPR and dCas9-KM proteins be phosphorylated by CDK1, and would this affect their activity?
4. Figure 6: It certainly appears that the majority of dCas9-VPR cells did not recover after Dox removal following 24 h treatment in the presence of the RO-3306 Cdk1 inhibitor, but it is not clear how fast the dCas9-VPR protein levels decay after Dox removal in this cell line, and if the dCas9-VPR protein remains present at significant levels then the authors have not really tested whether elimination of the dCas9-VPR protein would allow a greater fraction of cells to recover.
5. Were similar perturbations in heterochromatin structure observed when levels of MSR RNA were manipulated using induced dCas9-VPR and dCas9-KM expression in other cell types, including cancer cells, such as BRCA1 mutant breast cancer cells, where MSR RNA levels are already high and a compensatory mechanism might have been selected for?

Reviewer #2

(Remarks to the Author)

In this manuscript entitled "Forced expression of MSR repeat transcripts breaks heterochromatin organization" by Ching et al, the authors establish a system to bring transcriptional activators and repressors to pericentric heterochromatin (PCH) in mouse cells to study the downstream consequences with respect to PCH structure, satellite transcription, mitotic defects and viability. The authors find moderate effects when recruiting a repressor to the (anyway not very active) PCH region, and they find more striking effects when recruiting an activator, which results in dispersion and hyperacetylation of PCH, accumulation of satellite transcripts, mitotic defects and reduced viability. The first part of these results, dispersion and hyperacetylation of PCH, has already been published by others a few years ago. Compared to this previous work, the authors take a closer look at the major satellite repeat (MSR) transcripts and claim that their forced expression and accumulation at PCH is responsible for the observed effects. Unfortunately, I find no piece of data that actually tries to distinguish between the possibilities that the observed effects are induced either (i) by forced expression/accumulation of MSR transcripts, as the authors claim, or (ii) by changes of the chromatin state at PCH, for example by increased histone hyperacetylation, recruitment of transcriptional coactivators, potential loss of linker histone H1, etc, which accompany dCas9-VPR recruitment and likely affect chromatin folding. Also, it is not entirely clear to me how this work relates to earlier findings about RNA-dependent localization of HP1 at PCH and RNA-driven LLPS of HP1. Although I appreciate that the experimental system presented here is elegant, the data are of high quality, and the question/hypotheses are interesting, the manuscript in its present form does in my opinion not present particularly novel insights, and the main claim is not sufficiently justified. I have listed my main concerns below, along with suggestions for revisions.

Major points

1. From the mechanistic side, I find it key to distinguish between effects caused by the accumulation of MSR transcripts at PCH (or formation of MSR RNA:DNA hybrids) and effects caused by an active chromatin state at PCH (or actively ongoing transcription itself). Otherwise, I do not see how it can be claimed that it is the forced expression of MSR transcripts that is the driver of the observed phenotype. Chromatin acetylation and linker histone H1 have long been known to affect chromatin folding, and they have also been recently shown to have a direct effect on chromatin self-association/phase separation (PMID 31543265). It is therefore quite likely that they play some role in heterochromatin dispersion, even if the effect of HDAC inhibitors is different.

I suggest that the authors conduct experiments in which (i) the MSR transcripts are brought to PCH without changing the chromatin state, and (ii) the chromatin state is changed without accumulating MSR transcripts. The authors should have multiple possibilities to accomplish this: MSR transcripts could be fused/hybridized to the guide-RNA of their dCas9 system, they could be recruited by attaching an MS2 coat protein to dCas9 and an MS2 loop to the MSR transcript, etc. There is a good chance that this will not trigger histone acetylation, at least not immediately. To change the chromatin state without accumulating MSR transcripts, transcription could be blocked with inhibitors (as the activator is anyway expressed only for a few hours, it should be possible to find conditions where this is possible), or the MSR transcripts could be degraded with RNase (expressed onconase, RNase A/H on permeabilized cells). It would also help to compare VPR, which is an extremely strong activator, to weaker activators that might have differently pronounced effects on transcription and chromatin state. In addition, it would be interesting to know the effect of MSR transcripts provided in trans (delivered for example by transfection).

2. From the physiological side, I am concerned that the dCas9-VPR system is quite artificial. dCas9 has a very strong affinity to DNA, and VPR is a very strong activator. As the authors point out, wildtype cells express some MSR transcripts, and the level is even higher in early stages of differentiation. These levels do not seem to be toxic. It would therefore be important (i) to compare the expression levels used here to the levels observed in different physiological/pathological situations, (ii) determine more quantitatively which MSR expression level is needed to induce which of the effects reported here, and (iii)

test the effect of MSR expression levels that are similar to those seen naturally during development or in cancer cells. In the current manuscript, the authors show various effects triggered by an extreme overexpression of MSR transcripts, but it is difficult to interpret this finding.

I suggest that the authors quantify the absolute MSR transcript levels in control cells and cells expressing the activator/repressor, and to do the same for the physiological/pathological states they refer to in their manuscript. In this context, it would be very interesting to know the numbers for MEFs, ESCs, and the Suv39h1/2-deficient counterparts that probably express more MSR transcripts. It seems very well possible that the overexpression levels achieved here with VPR are orders of magnitude beyond the levels seen in these more natural situations, where strong activators that target PCH are probably absent. In this case, it would be helpful to adjust the dox concentration to study more realistic MSR expression levels, which would actually be a strength of the system the authors have established (with the caveat that MSR transcript levels are already increased without dox, according to Fig. 1d). This would also be interesting because it would allow addressing the question if the effects on PCH structure and chromatin state are connected to mitotic defects and cell viability, as lower VPR levels might induce some of the effects but not the others so that different aspects could be decoupled.

I find this point crucial because in a natural setting, where VPR-like activators are most probably absent, cells might not have to take special care to avoid extremely high MSR expression levels, but just decorate PCH with nucleosomes to keep it sufficiently repressed.

3. I am wondering how the present manuscript relates to previous work that has shown that an RNA component is required to retain HP1 at PCH and that has also proposed that Suv39 is recruited by RNAs (Almouzni lab and the last author's own work). In Fig. 2c, HP1 is retained at PCH although no MSR transcripts are present, and in Fig. S1d, Suv39h2 is retained at PCH in a similar situation where RNAs are degraded. I miss a discussion that explains what the authors think about these results. Is the conclusion that HP1 and Suv39h2 do not require MSR transcripts for localization to PCH, but other transcripts? And that these other transcripts are not fully degraded by onconase? Or that RNase A addition to permeabilized cells induces indirect effects that lead to HP1 loss? It would be helpful to extend this part.

4. It has been proposed that HP1 undergoes LLPS in the presence of MSR transcripts (PMID 35725842). This would favor round-shaped PCH foci, while the authors observe here the opposite when increasing MSR transcript expression with dCas9-VPR, namely the dispersion of round-shaped PCH, in agreement with previous work (PMID 32101700). It would be helpful if the authors could discuss this result. In this context, the absolute quantification of MSR transcript numbers I suggested above, as opposed to the relative numbers currently presented in the manuscript, would be very helpful. One possibility is that the level of MSR transcripts is below that needed to drive LLPS of HP1, and this would be an important information that the authors should be able to easily provide with their system (based on their FISH/qPCR/RNA-seq data).

Minor points

- The RNA-seq experiments are not fully exploited: Which genes are up-/down-regulated upon dCas9-VPR expressions? Are there any enriched GO terms or any other interesting findings? Are apoptotic markers upregulated?
- The authors show that the dCas9-VPR cell line shows a 50-60x upregulation of MSR transcripts without dox. As these cells grow, I assume that their viability is not compromised? This would already give a hint that the effects observed here require drastic MSR overexpression. What is the PCH morphology of these cells?
- The authors write a whole section in the Results about dCas9-VPR inducing histone hyperacetylation and recruitment of phosphorylated RNA polymerase II, without referencing an earlier paper that has presented very similar results (see Fig. 7H and Fig. S7 in PMID 32101700 for H3K27ac and RNAPII S5P stains). I think it would be fair to do so.

Reviewer #3

(Remarks to the Author)

In this manuscript the authors examine the role of major satellite RNA in heterochromatin organisation and integrity. This is a question of fundamental importance, towards the understanding how heterochromatin is organised and contributes to accurate chromosomal inheritance at mitosis. Establishing an inducible epigenetic activation and repression system targeting major satellite sequences, they achieve considerable modulation of major satellite transcription in MEF cells. They go on to demonstrate that both modulations lead to alterations in heterochromatin organisation, while forced overexpression of major satellites causes chromosome segregation errors and severely impacts cell viability.

The inducible system to perturb major satellite expression with dCas9 represents an exciting technical advance and improves upon previously published systems that have modulated major satellite expression. However, the biological findings reported here are not ground-breaking, as major satellite RNA is known to be important for heterochromatin organisation both in ES cells and early mammalian embryos (e.g. Lopes Novo et al., Nat. Comm. 2022, Erdel et al., Mol. Cell 2020, Casanova et al Cell Reports 2013). The novelty of this phenotype in MEF cells represents a fairly incremental advance in knowledge. It has also been demonstrated that depletion of major satellite RNA results in chromosomal instability (Lopes Novo et al). Here such a phenotype is found after strong activation of major satellite expression, although the physiological implications of this massive overexpression are not clear (see general comment 2 below). Although the technology used here is impressive, the authors unfortunately still cannot rule out effects of recruitment of epigenetic modifiers themselves, rather than major satellite transcription itself, as causal for the phenotypes observed, in this reviewer's

opinion (see general comment 3). With these and other caveats specified below in mind, I unfortunately cannot currently recommend publication of this manuscript in Nature Communications.

Major Comments

General comments

1. There is an absence throughout the manuscript of appropriate reporting of biological reproducibility, which is a foundation of experimental science. Please report the number of independent experiments performed as well as individual samples throughout.
2. A major concern is the leakiness of MSR-dCas9-Activator in -dox control and thus whether this can be used as an appropriate control throughout the manuscript. Although not detected by Western blot (Figure 1b), a considerable increase (50x) in MajSat RNA levels are observed in MSR-dCas9-Activator -dox conditions (Figure 1d). I question whether this condition is really 'uninduced' and thus is it an appropriate control? The authors need to compare phenotypes of MSR-dCas9-Activator -dox conditions to MSR-dCas9-Control throughout, to determine whether this 50x induction leads to changes in the phenotypes observed (e.g. Figure 2a). If not, why is 50x increase in transcription not sufficient but a 4000x fold increase in expression over unmodified cells is necessary? For example, if a 50-fold increase in MajSat transcript levels is not sufficient to cause chromosome segregation or apoptosis, what is the physiological relevance of a 4000-fold increase?
3. The authors use dCas9 fused to KRAB-MeCP2 to induce repression and observe an aggregation of MajSat regions (Figure 2). However, how can the authors exclude that this effect is simply due to targeting of the KRAB-MeCP2 effectors to Major Satellite rather than via decreased levels of MajSat RNA? Similar concerns are pertinent for the Activator (VPR), which may lead to decondensation independently of transcriptional changes.

Specific Comments

Figure 2

In addition to sphericity and volume in Figure 2a, number of segmented DAPI-dense regions should be reported, as the authors claim a decreased number of regions in MSR-dCas9-repressor conditions.

How robust are the measurements of sphericity, volume and number of segmented MSR-foci to segmentation method. How were thresholds chosen for segmentation? If thresholds are modified are the reported effects consistent?

Figure 2b – why is the data grouped in such a way? Please represent the data in a clearer way without grouping and provide appropriate statistical tests.

Figure 2c and Supplementary Figure 3 – are the presented linescans representative? Of how many images? How many independent experiments?

Figure 4

The authors do not mention the localisation of MajSat RNA in the control condition. Are they also associated with chromocentres? Does their MajSat RNA FISH data agree with previous research in ES cells (Lopes Novo et al). If not, why is only overexpressed MajSat associated with DAPI-rich regions? The authors should discuss the implications of this.

Data should be presented also in -Dox conditions (Figure 4a).

Figure 5

Chromosome segregation abnormalities shown in Figure 5c should be quantified with appropriated N numbers and statistical testing.

Figure 6

Quantification of the phenotypes referenced regarding Figure 6b should be provided.

Discussion

The lack of a strong phenotype in terms of chromosomal instability upon major satellite depletion differs to a previous study in ES cells (Lopes Novo). This should be discussed.

The authors argue that 'MSR RNA output, rather than the transcription process per se, is a crucial determinant in directing the structural organization of pericentric heterochromatin'. How do they discriminate between these possibilities in this manuscript? I don't see any evidence to support this statement, except for the TALE-Onconase experiments in Supplementary Figure 1, which had modest effects.

Minor Comments

Regarding Figure 3c, reference should be made to previous research reporting similar findings (Erdel et al., Mol. Cell 2020). Please provide the full sequence of MajSat gRNA.

Change scale of L1Md_A plot in Figure 1d

Version 1:

Reviewer comments:

Reviewer #1

(Remarks to the Author)

The authors have addressed the reviewers' comments as best they can by including a significant amount of new experimental data in Figures 4 and 5 and Figure S6. Their new data establishing the threshold level of MSR RNA expression required for disruption of heterochromatin foci in MEFs nicely explain why the relatively high basal level of MSR RNA in the MSR-dCas9-Activator cells in the absence of DOX was insufficient to cause disruption of foci, and also why the relatively small physiological increase in MSR RNA level observed in 2-cell embryos drives chromocenter formation rather than loss of foci. Their new data with RNase treatment, DRB inhibition of RNA Pol II activity, and A-485 inhibition of p300/CBP HAT activity suggest that MSR RNA expression and a transcriptionally hyperactive chromatin state are needed for the observed effects of MSR-dCas9-Activator driven MSR transcription in reducing heterochromatin condensates. Ultimately, however, more will need to be done to define the precise role of MSR RNA in the dynamics of pericentric heterochromatin foci formation and disassembly at a mechanistic level

Reviewer #2

(Remarks to the Author)

The authors have thoroughly revised and strengthened their manuscript entitled "Forced expression of MSR repeat transcripts above a threshold limit breaks heterochromatin organisation". They have established a (relative) threshold of MSR transcripts above which heterochromatin foci are dispersed, and they have conducted additional experiments to test the influence of MSR transcription and chromatin state on heterochromatin organization. I have some mostly minor comments:

1. The authors conclude that both RNA output and a transcriptionally hyperactive chromatin state influence heterochromatin organization. I wonder if the title of the manuscript could be adjusted to convey this conclusion more clearly.
2. The authors determine a threshold level of MSR transcripts above which heterochromatin is dispersed. According to Fig. 4d, mammary gland cells have higher MSR transcript levels (50,000 normalized reads) than 6KO cells (20,000 normalized reads) that have dispersed heterochromatin. Presumably mammary gland cells contain focal heterochromatin structures and do not have the problems associated with dispersed heterochromatin that the authors describe here, which speaks against an absolute threshold. I suppose that the authors indicate relative thresholds (300-fold instead of a number of normalized counts) for this reason. Could the authors speculate how the cell type-dependence of this threshold can be explained? Are there any obvious changes in heterochromatin composition in mammary gland cells?
3. I find it difficult to reconcile the discussion about RNA-HP1 condensates with the authors' data and the data in ref. 10. An increase of MSR transcript could indeed saturate HP1 RNA binding and disfavor inter-nucleosomal protein-protein interactions, explaining the switch from aggregated to dispersed heterochromatin. However, Suv39h double-null MEFs where most HP1 is displaced from heterochromatin are fairly similar to wildtype MEFs: They have focal heterochromatin structures that become dispersed when dCas9-VPR is recruited (ref 10). This seems to suggest that HP1 does not have a strong effect on the aggregation of heterochromatin foci and their VPR-dependent dispersion.

Reviewer #3

(Remarks to the Author)

While the authors have performed a significant amount of work, with the addition of Figures 4 and 5, unfortunately I am still not in support of publication as my major comments have not been fully addressed.

Major Comments

General comments

1. There is an absence throughout the manuscript of appropriate reporting of biological reproducibility, which is a foundation of experimental science. Please report the number of independent experiments performed as well as individual samples throughout.

We have carefully revised the Figures and Figure legends and now provide statistical analyses, number of independent experiments and n-numbers of individual cells or images analysed. This is now specified in the Figure legends.

This point is still not addressed. There is still a lack of reporting of number of independent experiments throughout the manuscript. For example, Figures 1b and c do not mention the number of times the experiment was repeated. This is also not addressed in the examples presented in the specific comments below.

2. A major concern is the leakiness of MSR-dCas9-Activator in -dox control and thus whether this can be used as an appropriate control throughout the manuscript. Although not detected by Western blot, a considerable increase (50x) in MajSat RNA levels are observed in MSR-dCas9-Activator -dox conditions (Figure 1d). I question whether this condition is really 'uninduced' and thus is it an appropriate control? The authors need to compare phenotypes of MSR-dCas9-Activator -dox conditions to MSR-dCas9-Control throughout, to determine whether this 50x induction leads to changes in the phenotypes observed (e.g. Figure 2a). If not, why is 50x increase in transcription not sufficient but a 4000x fold increase in expression over unmodified cells is necessary? For example, if a 50-fold increase in MajSat transcript levels is not sufficient to cause chromosome segregation or apoptosis, what is the physiological relevance of a 4000-fold increase?

Our new data on the definition of a threshold limit for forced MSR expression in disrupting heterochromatin (new Figure 4)

(see also response to reviewer 1 and reviewer 2) addresses these questions and can explain that 40-fold increased MSR transcript levels are significantly below the threshold limit (300-fold upregulation) and do not induce dispersion of DAPI-dense foci. In addition, we quantified mitotic defects and chromosome mis-segregation in a new supplementary Figure S7. Mitotic defects are modestly increased by 2-fold in MSR-dCas9- Activator(-dox) as compared to MSR-dCas9-Control(-dox) but are >9-fold higher in MSR dCas9-Activator(+dox). These data support our conclusions that increasing levels of MSR RNA manifest with higher and more frequent mitotic defects. There are no cellular pathways were dysregulated between MSR-dCas9-Activator(-dox) and MSR-dCas9-Control(-dox).

I applaud the authors for attempting to address a threshold limit of MSR transcripts for heterochromatin integrity, in a new Figure 4, which goes does address the leakiness concern of the uninduced condition. However, the following points limit my enthusiasm for this.

- In Figure 4a (and Supplementary Figure 6a) the number of experiments is not reported.
- Statistical testing is not shown for Figure 4c (and Supplementary Figure 6c). Thus, the conclusion that dispersion is observed at 2 ng/ml dox for example, is not supported. As this is the basis for setting the threshold of MSR transcripts, I am not in support of publication of this conclusion.
- In many new panels they perform a counting of dispersed vs undispersed heterochromatin phenotypes. As far as I can tell the determination of how dispersed vs undispersed were defined and how was the data binarized in such a way? Was this done manually? If so, was this done blind? How was blinding achieved? Quantification of dispersion should be provided, similarly to Figure 2, with volume and number of domains. This also applies to Supplementary Figure 6d and Figure 5c/f.
- I do not follow the calculations for fold change in MSR RNA sequences (lines 323-327). They describe 220,000 normalized reads in the induced condition, which compared to 600 normalized reads in uninduced control, represents a 367-fold increase, not >470-fold as they report. Similarly, compared to 2000 normalized reads in the uninduced MSR-dCas9-Activator condition, my calculation gives a 110-fold increase not >150-fold as they describe.
- If a threshold of 300-fold MSR upregulation is necessary for dispersion of heterochromatin, why is a complete collapse observed in 6KO MEF cells, where only an 80-fold MSR upregulation is observed compared to wt cells?
- How specific is this threshold for MEF cells? It is not clear how this threshold would apply to other cell types. A cell-type threshold dependence could exist for a number of reasons, including different organisation and proteome of chromocenter-associated heterochromatin in different cell types and different recruitment of activators. At least, the authors should specify that this specific threshold would apply to MEF cells and cannot necessarily crossover to other cell types. Thus, the discussion of the physiological relevance and the direct application of this threshold to other cell types in the relevant section in the discussion should be toned down.

3. The authors use dCas9 fused to KRAB-MeCP2 to induce repression and observe an aggregation of MajSat regions (Figure 2). However, how can the authors exclude that this effect is due simply to targeting of the KRAB-MeCP2 effectors to Major Satellite rather than via decreased level of MajSat RNA? Similar concerns are pertinent for the Activator (VPR), which may lead to decondensation independently of transcriptional changes.

This cannot be excluded and is indeed clearly stated (“...heterochromatic regions aggregate. This could primarily be caused by increased recruitment of components of the silencing machinery and protein-based coalescence...”) in the first paragraph of the discussion. As already explained in response to reviewer 2, approaches to uncouple RNA output from an active chromatin state has been a long-standing challenge in the field. We engaged in several new approaches, such as RNase A treatment of permeabilized cells (new supplementary Figure 6d), inhibition of RNA Pol II transcription (new Figure 5a-c) and reducing histone acetylation (new Figure 5d-g). From these new approaches, we conclude that both RNA output and a transcriptionally hyperactive chromatin state influence heterochromatin organization. These are important new data sets that we have added and also present and discuss accordingly.

Apologies for missing this point in the discussion previously. I commend the authors for attempting to address this difficult point using three different approaches. However, I have a number of concerns about the data:

- The quantification of dispersed vs undispersed in Figure 5c/f is not clearly described and statistical testing is not shown (see 2nd and 3rd points above).
- The results in Figures 5f and 5g appear inconsistent. In Figure 5f a 5 μ M concentration of A-485 leads to 30% dispersion. In Figure 5g, under comparable conditions 75% dispersion is observed (without DRB). The only difference as far as I can tell is that Dox is applied for only 1h in 5g compared to 6h in 5f. However, this should decrease the amount of MSR transcript produced, and thus fewer dispersed chromocentres should be observed.
- The results of these experiments are not conclusive, even ignoring the caveats described above. Unfortunately, the acetyltransferase inhibitor leads to a decrease in transcription of MSR and thus an effect of acetylation independently from MSR transcripts cannot be ascribed. The transcriptional inhibition or RNaseA treatment do not rescue heterochromatin dispersion (although it is not clear if either of these differences are reproducible or statistically significant), suggesting that transcription or the RNA are not involved in the dispersion. The only evidence that the MSR transcript itself may be involved in the dispersion is in Figure 5g, which shows a difference +/- DRB only at 5 μ M A-485, the results of which are not consistent with Figure 5f, as mentioned above. Thus, it is still not clear to what extent the MSR transcripts vs the recruitment of activators contributes to heterochromatin integrity.

Specific Comments

Figure 2

In addition to Sphericity and Volume in Figure 2a, number of segmented DAPI-dense regions should be reported, as the authors claim a decreased number of regions in MSR-dCas9-repressor conditions.

How robust are the measurements of sphericity, volume and number of segmented MSR-foci to segmentation method. How were thresholds chosen for segmentation? If thresholds are modified are the reported effects consistent?

We did not count the number of DAPI-dense regions, but rather analysed the number of CREST puncta per DAPI-dense region, which shows higher number of CREST puncta per DAPI-dense region in the MSR-dCas9-Repressor MEF cells. These data have been moved to supplementary Figure 3a, where we now also provide statistical testing. The measurements of sphericity and volume of DAPI-dense regions are robust and maintain the observed differences when threshold levels are modified.

In lines 188-189 the authors still claim that heterochromatic regions are reduced in number in MSR-Cas9-Repressor conditions. The data is still not in the manuscript to support this statement.

Figure 2b – why is the data grouped in such a way? Please represent the data in a clearer way without grouping and provide appropriate statistical tests.

The CREST data have been moved to supplementary Figure 3a, where we now also provide statistical testing.

The data is now presented more clearly and the number of quantified regions are described. However, they still do not describe the number of independent experiments.

Figure 2c and Supplementary Figure 3 – are the presented linescans representative? Of how many images? How many independent experiments?

The linescans shown are derived from the displayed images. Images are representative from three biological replicates. The number of cells analysed are indicated in the Figure legends.

The number of independent experiments is still not reported in the manuscript.

Figure 4

The authors do not mention the localisation of MajSat RNA in the control condition. Are they also associated with chromocentres? Does their MajSat RNA FISH data agree with previous research in ES cells (Lopes Novo et al). Data should be presented also in -Dox conditions (Figure 4a).

If not, why is only overexpressed MajSat associated with DAPI-rich regions? The authors should discuss the implications of this.

MSR transcripts reach higher levels in mESC as compared to MEF cells. MSR RNA has been reported to largely remain associated with pericentric heterochromatin in mouse cells and is consistent with previous reports, including our work in MEF cells (Bulut-Karslioglu et al., 2012) and with Lopes Novo et al., 2022 in mESC. Signals for MSR transcripts in the MSR-dCas9- Control(+dox) are low, since exposure times for imaging were adjusted to the high signals for MSR transcripts in the MSR-dCas9-Activator(+dox). We also did the MSR RNA-FISH analyses in the uninduced conditions, where we did not observe differences for the MSR RNA signals in MSR-dCas9-Control, MSR-dCas9-Repressor and MSR-dCas9-Activator MEF cells (data not shown).

The faint MSR AS and S signal in control conditions is not localized to pericentric heterochromatin, as they claim above. There appears to be low level nuclear signal in both conditions, that is absent in the Cas9-Repressor conditions, suggesting that this faint signal is specific. This point is not sufficiently addressed.

Figure 5

Chromosome segregation abnormalities shown in Figure 5c should be quantified with appropriated N numbers and statistical testing.

This is now added as new supplementary Figure S7

The quantification has now been added. However, the number of independent experiments is still not reported in the manuscript.

Figure 6

Quantification to the phenotypes referenced in Figure 6b should be provided.

More than 80% of MSR-dCas9-Activator MEF cells 24 hrs post-induction show early to late apoptosis, as quantified by flow cytometry in Figure 7a (right panel).

Number of independent experiments are not described.

Discussion

The lack of a strong phenotype in terms of chromosomal instability upon major satellite depletion differs to a previous study in ES cells (Lopes Novo). This should be discussed.

We discuss the differences in number and size of heterochromatic foci between mouse ES cells (Lopes Novo et al., 2022), serum-free mESC and MEF cells (our study). We have extended this discussion point and now also refer to the 50% reduced level of MSR transcripts in the (Lopes Novo et al., 2022) paper.

This has been addressed.

The authors argue that 'MSR RNA output, rather than the transcription process per se, is a crucial determinant in directing the structural organization of pericentric heterochromatin'. How do they discriminate between these possibilities in this manuscript? I don't see any evidence to support this statement, except for the TALE-Onconase experiments in Supplementary Figure 1, which had modest effects.

This has indeed been the most challenging part of the study and of the revision. As explained above, we addressed this question with several new approaches, such as RNase A treatment of permeabilised cells (new supplementary Figure 6d), inhibition of RNA Pol II transcription (new Figure 5a-c) and reducing histone acetylation (new Figure 5d-g). From these new approaches, we conclude that both RNA output and a transcriptionally hyperactive chromatin state influence heterochromatin organisation. These are important new data sets that we have added and also present and discuss accordingly.

See my response to major comment 3.

Version 2:

Reviewer comments:

Reviewer #2

(Remarks to the Author)

The authors have replied to my questions in the last revision round and I am fine with their answers.

Overall, it is a bit unfortunate that it remains unclear if the contribution of the MSR transcripts themselves or that of the active chromatin state is stronger in breaking heterochromatin. As I had suggested in the first revision round, I thought that this could have been tested by directly tethering the MSR transcripts (produced elsewhere) instead of recruiting the transcriptional Activator (to transcribe endogenous MSRs). In any case, I agree with the authors that their data suggest that both the MSR transcripts and the active chromatin state affect heterochromatin organisation. This model suggests that there is a threshold limit for MSR transcripts, which receives lots of attention in the manuscript, but also a threshold limit for active histone marks (or for components of the transcription machinery that accumulate over MSRs), above which heterochromatin organisation is disrupted. This aspect could be presented in a more balanced way, at least in the Discussion. The existence of another threshold, one for active histone marks and activator proteins, could also explain that the threshold for MSR transcripts is cell type-dependent, as it is not the only determinant of heterochromatin organisation.

Reviewer #3

(Remarks to the Author)

The authors argue based on reputation that their data is robust: 'We strongly disagree with this exaggerated criticism, as we have a long-standing record in generating high-quality and reproducible data sets.' This is unfounded as science is assessed by individual merit rather than reputation. As many of my relatively simple requests were not addressed in the previous round of review (even though the authors claimed they were) I naturally requested them again and thus I do not believe my criticism is exaggerated. Many of my points have now been addressed, but I still have some outstanding points, most importantly pertaining to the definition of the 300-fold threshold, an important conclusion of the manuscript.

1. I thank the authors for now providing reproducibility reporting information.

2. The experimental variability (error bars across independent experiments) and statistical testing are still not shown for Figure 4c and Supplementary Figure 6c. Thus, how can they conclude that 2ng/ml dox (which equates to 300-fold induction) induces dispersion? The apparent increase in cells with dispersed heterochromatin observed at 2ng/ml could be within experimental variability. Moreover, I am still not convinced by the manual quantification of dispersed vs undispersed foci. No blinding is described, and thus the authors cannot exclude subconscious bias in this counting. As far as I can tell a more unbiased quantification of foci number and volume is not provided, as requested.

3. No explanation is provided for the inconsistent results between Figure 5f and g.

Specific comments

Figure 4

The authors still claim that the signal of the MSR RNA FISH in control conditions overlaps with DAPI and Cas9 foci. This is clearly not the case, especially in the image shown with the ASR probe.

All other comments are addressed.

Version 3:

Reviewer comments:

Reviewer #2

(Remarks to the Author)

The authors have addressed my remaining point in the Discussion and I support publication.

Reviewer #3

(Remarks to the Author)

I thank the authors for adding the statistical testing to Figure 4c. Please could the authors also add error bars as requested or individual data points to show the experimental variability. This also applies to similar quantification in Figures 5c and f and Supplementary Figure 6c.

All other points have been addressed.

REVIEWER COMMENTS NCOMMS 24-09802

Reviewer #1 (Remarks to the Author):

Here the authors set out to determine the consequences for heterochromatin formation of inducibly enforcing over- or under-expression of the major satellite repeat (MSR) RNA in mouse embryo fibroblasts (MEFs). For this purpose, they transduced MEFs using the PiggyBac transposon expression system and established cell lines in which the dCas9-VPR transcriptional activator and the dCas9-KM transcriptional repressor proteins could be inducibly expressed and targeted to MSR by a single gRNA that is constitutively expressed from the U6 promoter cassette integrated into the TRE-dCas9 plasmids. They showed that the Dox-induced dCas9 fusion proteins were targeted to pericentric heterochromatin, as monitored by DAPI and dCas9 staining, and that induction of the dCas9-VPR fusion protein led to a 4,000-fold increase in MSR-RNA and rapid cell death, whereas induction of the dCas9-KM fusion protein strongly reduced the level of MSR-RNA, with expression of the control dCas9 protein having no effect. A time course of the responses of cells to induction analyzed by staining with CREST anti-centromere antibodies showed that dCas9-VPR induced expression of MSR-RNA resulted in rapid dispersion of heterochromatin foci that still exhibited HP1 β , HP1 γ and HMGA1 co-localization, whereas induction of dCas9-KM caused delayed aggregation of heterochromatin. They also found that the levels of histone H3 pan-Ac and recruited RNA Pol II were elevated in dCas9-VPR activator domains. Analysis by immunoRNA-FISH using strand-specific MSR probes showed that both the levels of both the forward and reverse MSR RNAs were elevated and remained localized to dCas9-VPR activator domains, and by probing a dot-blot with an antibody that detects RNA:DNA hybrids and carrying out RDIP they showed that MSR DNA-RNA hybrids were formed, validating the existence of RNA-DNA hybrids based on their disappearance upon prior treatment with RNase H. Finally, by using the RO-3306 CDK1 inhibitor to arrest cells in G2 followed by release into M phase, they showed that induced expression of dCas9-VPR delayed mitosis, as monitored by decreased pS10 H3 levels, and also led to chromosome mis-segregation, with a high frequency of lagging chromosomes. Prolonged induction of dCas9-VPR expression elicited MEF cell death and apoptosis, which were shown to be partially due to the observed mitotic defects, with other mechanisms also being responsible.

The use of inducible dCas9-activator and dCas9-repressor protein expression to regulate the expression levels of the major satellite repeat RNA in cells is an elegant approach to deciphering the function of MSR-RNA in pericentric heterochromatin organization. The experimental flow is logical and the conclusions are largely justified by the data, and support an important role for MSR transcripts in the structural organization of pericentric heterochromatin. Overall, these are interesting findings, although the mechanisms underlying the observed changes in heterochromatin structure have not been fully elucidated, and it would be interesting to know whether MSR RNA-scaffolded LLPS formation also occurs in the MEF cell system, as was reported for mESCs (ref. 11).

Points: 1. Figure 1: How stable are the dCas-fusion proteins, i.e. do they continue to accumulate following induction or do they reach a steady state level where of synthesis and turnover are balanced during these experiments? A schematic of the genomic organization of the MSR repeat and minor satellite region showing where potential TSS's map and where the gRNA target sites are localized would be helpful. In panel D, the level of MSR RNA was apparently 40-fold higher in the dCas9VPR cells than in the dCas9 control cells, even though according to the blots in panel A there does not appear to have been any dCas9-VPR protein in the uninduced cells. A constitutive 40-fold increase in MSR RNA level over that in control cells might have been expected to elicit some heterochromatin-dependent phenotypic changes. What is observed if the RNA expression patterns of the dCas9-VPR (-Dox cells) are compared to the dCas9 (-Dox) cells (panel E), i.e are there induced transcripts? Further discussion of these issues is warranted.

The expression and stability of the MSR-dCas9-effector components was analysed post-induction via Western blot and MSR-dCas9-effectors protein levels decreased (see Supplementary Figure 8a).

The genomic organization of MSR repeat arrays and potential TSS have been described in (Bulut-Karslioglu et al., 2012) and this is cited. We also add the DNA sequence of the MSR consensus sequence and show the gRNA target site in Figure 1a.

Our new definition of a threshold limit for forced MSR expression in disrupting heterochromatin (new Figure 4, see also below) addresses these questions and can explain that 40-fold increased MSR transcript levels are significantly below the threshold limit (300-fold upregulation) and do not induce dispersion of DAPI-dense foci.

2. Figure 4: Given the huge increase in the levels of MSR RNA upon induction of the dCas9-VPR activator it seems unlikely that that much of it can form DNA:RNA hybrids - have the authors estimated the copy numbers per cell of the forward and reverse MSR RNAs upon induction compared to the number DNA copies of the MSR repeats? Does the RNase A sensitivity of the FISH signals in the dCas9-VPR activator panels mean that the forward and reverse strands RNA products are not forming dsRNA, or is dsRNA not denatured under the conditions used for RNA-FISH, and therefore not detectable. In this connection, can the authors test whether pretreatment with a dsRNase, such as RNase III, affects the total MSR RNA level?

3.6% of mouse genomic DNA are major satellite repeats (Waterston et al., 2002). As the consensus MSR unit is 234 bp, this accounts for around 385,000 MSR copies in the mouse genome. However, it has been estimated that only < 15% of the MSR copies preserve high sequence identity with the MSR consensus sequence and maintain transcriptional competence (Bulut-Karslioglu et al., 2012). While we have not directly analyzed copy numbers for forward and reverse MSR transcripts, our HiSeq RNA sequencing indicates between 200-600 normalized read counts for MSR sequences at uninduced and wt conditions. This number is significantly increased to > 220,000 normalized read counts for MSR sequences in the MSR-dCas9-Activator (+dox) MEF cells. We now provide these numbers of normalized read counts for MSR sequences in uninduced and induced conditions in a comparative analysis shown in new Figure 4d.

In our previous work (Velazquez-Camacho et al., 2017), the nature of an RNA-nucleosome scaffold has been analyzed and MSR RNA was shown to form RNA:DNA hybrids with MSR DNA or ssRNA. Nonetheless, a recent report describing Dicer-deficient mESC (Gutbrod et al., 2022) has indicated that MSR transcripts are highly elevated in the absence of Dicer (an RNase III enzyme). We have referred to this in the revised discussion.

3. Figure 5: Can the dCas9-VPR and dCas9-KM proteins be phosphorylated by CDK1, and would this affect their activity?

We have not addressed this question.

4. Figure 6: It certainly appears that the majority of dCas9-VPR cells did not recover after Dox removal following 24 h treatment in the presence of the RO-3306 Cdk1 inhibitor, but it is not clear how fast the dCas9-VPR protein levels decay after Dox removal in this cell line, and if the dCas9-VPR protein remains present at significant levels then the authors have not really tested whether elimination of the dCas9-VPR protein would allow a greater fraction of cells to recover.

The stability of the MSR-dCas9-effector components has been analysed after removal of doxycycline (new supplementary Figure 8). While MSR-dCas9-effector components progressively decline, forced MSR transcript levels in the MSR-dCas9-Activator MEF cells remain in high abundance, even 24 hrs post-induction, and do not decline.

5. Were similar perturbations in heterochromatin structure observed when levels of MSR RNA were manipulated using induced dCas9-VPR and dCas9-KM expression in other cell types, including cancer cells, such as BRCA1 mutant breast cancer cells, where MSR RNA levels are already high and a compensatory mechanism might have been selected for?

Rather than transferring the inducible MSR-dCas9-effector components into other cell types, we have invested into defining a threshold limit for the forced expression of MSR transcripts. We attenuated MSR-dCas9-Activator induction with low dose titrations of doxycycline (new Figure 4a-c) and also through a time course (new supplementary Figure 6a-c). The data reveal that dispersion of heterochromatic foci requires a threshold limit of >300-fold upregulation of MSR transcripts. Full dispersion of heterochromatic foci plateaus at 4,000-fold upregulation of MSR transcripts. The identification of a threshold limit for forced MSR upregulation in dispersing heterochromatic foci is an important new result, as it can now be related to other studies describing derepression of MSR transcripts in both physiological and pathological settings. For example, in mouse 2 cell stage embryos, a 'burst' of MSR transcription elevates MSR transcripts by around 50-100 fold (Probst et al., 2010) and is required for chromocenter formation. This increase is below the threshold level for forced MSR upregulation in dispersing heterochromatic foci. Further, in BRCA1-deficient mouse brain tissue (Zhu et al., 2011) and in mouse breast cancer tissues (Zhu et al. 2018), MSR transcript levels are already highly elevated and are above the threshold limit (new Figure 4d). We have added these new data in defining a threshold limit for forced MSR transcript levels as new Figure 4 and also present and discuss them accordingly.

Reviewer #2 (Remarks to the Author):

In this manuscript entitled "Forced expression of MSR repeat transcripts breaks heterochromatin organization" by Ching et al, the authors establish a system to bring transcriptional activators and repressors to pericentric heterochromatin (PCH) in mouse cells to study the downstream consequences with respect to PCH structure, satellite transcription, mitotic defects and viability. The authors find moderate effects when recruiting a repressor to the (anyway not very active) PCH region, and they find more striking effects when recruiting an activator, which results in dispersion and hyperacetylation of PCH, accumulation of satellite transcripts, mitotic defects and reduced viability. The first part of these results, dispersion and hyperacetylation of PCH, has already been published by others a few years ago. Compared to this previous work, the authors take a closer look at the major satellite repeat (MSR) transcripts and claim that their forced expression and accumulation at PCH is responsible for the observed effects. Unfortunately, I find no piece of data that actually tries to distinguish between the possibilities that the observed effects are induced either (i) by forced expression/accumulation of MSR transcripts, as the authors claim, or (ii) by changes of the chromatin state at PCH, for example by increased histone hyperacetylation, recruitment of transcriptional coactivators, potential loss of linker histone H1, etc, which accompany dCas9-VPR recruitment and likely affect chromatin folding. Also, it is not entirely clear to me how this work relates to earlier findings about RNA-dependent localization of HP1 at PCH and RNA-driven LLPS of HP1. Although I appreciate that the experimental system presented here is elegant, the data are of high quality, and the question/hypotheses are interesting, the manuscript in its present form does in my opinion not present particularly novel insights, and the main claim is not sufficiently justified. I have listed my main concerns below, along with suggestions for revisions.

Major points

1. From the mechanistic side, I find it key to distinguish between effects caused by the

accumulation of MSR transcripts at PCH (or formation of MSR RNA:DNA hybrids) and effects caused by an active chromatin state at PCH (or actively ongoing transcription itself). Otherwise, I do not see how it can be claimed that it is the forced expression of MSR transcripts that is the driver of the observed phenotype. Chromatin acetylation and linker histone H1 have long been known to affect chromatin folding, and they have also been recently shown to have a direct effect on chromatin self-association/phase separation (PMID 31543265) (Gibson et al. Cell 2019) It is therefore quite likely that they play some role in heterochromatin dispersion, even if the effect of HDAC inhibitors is different.

I suggest that the authors conduct experiments in which (i) the MSR transcripts are brought to PCH without changing the chromatin state, and (ii) the chromatin state is changed without accumulating MSR transcripts. The authors should have multiple possibilities to accomplish this: MSR transcripts could be fused/hybridized to the guide-RNA of their dCas9 system, they could be recruited by attaching an MS2 coat protein to dCas9 and an MS2 loop to the MSR transcript, etc. There is a good chance that this will not trigger histone acetylation, at least not immediately. To change the chromatin state without accumulating MSR transcripts, transcription could be blocked with inhibitors (as the activator is anyway expressed only for a few hours, it should be possible to find conditions where this is possible), or the MSR transcripts could be degraded with RNase (expressed onconase, RNase A/H on permeabilized cells). It would also help to compare VPR, which is an extremely strong activator, to weaker activators that might have differently pronounced effects on transcription and chromatin state. In addition, it would be interesting to know the effect of MSR transcripts provided in trans (delivered for example by transfection).

Approaches to uncouple RNA output from an active chromatin state has been a long-standing challenge in the field. We thank this referee for the thoughtful comments and we engaged in most of the suggested approaches, such as RNase A treatment of permeabilised cells, inhibition of RNA Pol II transcription and reducing histone acetylation. First, we optimised protocols for permeabilisation of agarose embedded cells, followed by incubation with RNase A. However, dispersed heterochromatic foci remain disrupted after RNase A treatment, although they lose HP1 association. We add these data as new supplementary Figure 6c. Second, we used DRB inhibition of RNA Pol II. Inhibition of RNA Pol II nearly abolished MSR transcripts in the MSR-dCas9-Activator samples, but only modestly regressed dispersion of heterochromatic foci. We add these data as new Figure 5a-c. Third, we applied an HAT inhibitor (A-485, targeting CBP and p300 HAT enzymes) to modulate an active chromatin state. Intriguingly, incubation of dCas9-Activator MEF cells with A-485 did prevent dispersion of heterochromatic foci. However, this HAT inhibition also resulted in a significant down-regulation of the forced MSR transcript levels to a near background level (new Figure 5d-f). Importantly, combined incubation with DRB and A-485 inhibitors further reduced the fraction of dCas9-Activator MEF cells with dispersed heterochromatin (new Figure 5g). Thus, both RNA output and a transcriptionally hyperactive chromatin state influence heterochromatin organization and these two principles do not appear that they could be uncoupled. These are important new data sets that we have added as a completely new Figure 5 and also present and discuss accordingly.

2. From the physiological side, I am concerned that the dCas9-VPR system is quite artificial. dCas9 has a very strong affinity to DNA, and VPR is a very strong activator. As the authors point out, wildtype cells express some MSR transcripts, and the level is even higher in early stages of differentiation. These levels do not seem to be toxic. It would therefore be important (i) to compare the expression levels used here to the levels observed in different physiological/pathological situations, (ii) determine more quantitatively which MSR expression level is needed to induce which of the effects reported here, and (iii) test the effect of MSR expression levels that are similar to those seen naturally during development or in cancer cells. In the current manuscript, the authors show various effects triggered by an extreme overexpression of MSR transcripts, but it is difficult to interpret this finding.

I suggest that the authors quantify the absolute MSR transcript levels in control cells and cells expressing the activator/repressor, and to do the same for the physiological/pathological states they refer to in their manuscript. In this context, it would be very interesting to know the numbers for MEFs, ESCs, and the Suv39h1/2-deficient counterparts that probably express more MSR transcripts. It seems very well possible that the overexpression levels achieved here with VPR are orders of magnitude beyond the levels seen in these more natural situations, where strong activators that target PCH are probably absent. In this case, it would be helpful to adjust the dox concentration to study more realistic MSR expression levels, which would actually be a strength of the system the authors have established (with the caveat that MSR transcript levels are already increased without dox, according to Fig. 1d). This would also be interesting because it would allow addressing the question if the effects on PCH structure and chromatin state are connected to mitotic defects and cell viability, as lower VPR levels might induce some of the effects but not the others so that different aspects could be decoupled.

I find this point crucial because in a natural setting, where VPR-like activators are most probably absent, cells might not have to take special care to avoid extremely high MSR expression levels, but just decorate PCH with nucleosomes to keep it sufficiently repressed.

We have now identified a threshold limit for forced upregulation of MSR transcripts. We attenuated MSR-dCas9-Activator induction with low dose titrations of doxycycline (new Figure 4a-c) and also through a time course (new supplementary Figure 6a-c). These data reveal that dispersion of heterochromatic foci starts within 2-3 hrs after MSR-dCas9-Activator induction and requires a threshold level of >300-fold upregulation of MSR transcripts. Full dispersion of heterochromatic foci plateaus at 4,000 fold upregulation of MSR transcripts. The identification of a threshold limit for forced MSR upregulation in dispersing heterochromatic foci is an important new result, as it can now be related to other studies describing derepression of MSR transcripts in both physiological and pathological settings. For example, in mouse 2 cell stage embryos, a 'burst' of MSR transcription elevates MSR transcripts by around 50-100 fold (Probst et al., 2010) and is required for chromocenter formation. This increase is below the threshold level for forced MSR upregulation in dispersing heterochromatic foci. In addition, we can now also explain why the 40-fold increased MSR levels in the MSR-dCas9-Activator(-dox) samples do not induce dispersion of heterochromatic foci. We have added these new data sets as new Figure 4 and new supplementary Figure 6a-c.

Rather than quantifying absolute MSR transcript levels, we have been using normalised read counts from our HiSeq RNA sequencing which allows for a comparative analysis and across distinct genetic backgrounds and different sequencing depths. In uninduced or wt conditions, between 200 to 500 normalised read counts for MSR sequences are found and this is between 5-10 fold below normalized read counts for housekeeping genes (e.g. Hprt). The number of normalised read counts for MSR sequences is significantly increased to >220,000 in the MSR-dCas9-Activator(+dox) MEF cells but only to 1,200 in Suv39h double-null MEF cells. We now provide these numbers of normalised read counts for MSR sequences in uninduced vs. induced conditions and in distinct genetic backgrounds (e.g. Suv39h double-null and also 6KO MEF cells) in a comparative analysis shown in new Figure 4d.

3. I am wondering how the present manuscript relates to previous work that has shown that an RNA component is required to retain HP1 at PCH and that has also proposed that Suv39 is recruited by RNAs (Almouzni lab and the last author's own work). In Fig. 2c, HP1 is retained at PCH although no MSR transcripts are present, and in Fig. S1d, Suv39h2 is retained at PCH in a similar situation where RNAs are degraded. I miss a discussion that explains what the authors think about these results. Is the conclusion that HP1 and Suv39h2 do not require MSR transcripts for localization to PCH, but other transcripts? And that these other transcripts are not fully degraded by onconase? Or that RNase A addition to permeabilized cells induces

indirect effects that lead to HP1 loss? It would be helpful to extend this part.

The new supplementary Figure 6d on RNase A treatment of permeabilised dCas9-Activator MEF cells shows loss of HP1 association, consistent with the early work by Maison et al., 2002. Suv39h1 and Suv39h2 enzymes appear less sensitive to RNase A treatment, although MSR transcripts contribute to their recruitment and stabilise their retention at heterochromatin (Velazquez-Camacho et al., 2017). We have clarified the RNA binding and RNase A sensitivity of HP1 in the revised discussion.

4. It has been proposed that HP1 undergoes LLPS in the presence of MSR transcripts (PMID 35725842) (Novo et al., 2022). This would favor round-shaped PCH foci, while the authors observe here the opposite when increasing MSR transcript expression with dCas9-VPR, namely the dispersion of round-shaped PCH, in agreement with previous work (PMID 32101700) (Erdel et al., 2020). It would be helpful if the authors could discuss this result. In this context, the absolute quantification of MSR transcript numbers I suggested above, as opposed to the relative numbers currently presented in the manuscript, would be very helpful. One possibility is that the level of MSR transcripts is below that needed to drive LLPS of HP1, and this would be an important information that the authors should be able to easily provide with their system (based on their FISH/qPCR/RNA-seq data).

The possible involvement of RNA-protein condensates for heterochromatin formation/stability can be interpreted through phase separation or with other biophysical or polymer models. We have already described the differences in number and size of heterochromatic foci between mouse ES cells (Novo et al., 2022), serum-free mESC and MEF cells (our study). We have extended this discussion point and now also refer to the 50% reduced level of MSR transcripts in the (Novo et al., 2022) paper.

Importantly, our new data sets on the definition of a threshold limit (new Figure 4), titration of MSR transcript levels (new Figure 4b and supplementary Figure 6b), RNase A sensitivity of HP1 localization to DAPI-dense regions (new supplementary Figure 6d) and the association of HP1 with dispersed A/T-rich MSR repeats (Figure 2b) now allowed for a completely revised and restructured discussion where we have added a new paragraph that focuses on the function of RNA in buffering liquid-like to solid-like (hetero)chromatin compartments.

Minor points

- The RNA-seq experiments are not fully exploited: Which genes are up-/down-regulated upon dCas9-VPR expressions? Are there any enriched GO terms or any other interesting findings? Are apoptotic markers upregulated?

We now provide gene expression and GO pathway analyses for MSR-dCas9-Activator MEF cells that show down-regulation of DNA repair and chromosome segregation control (new Figure 7b-c).

- The authors show that the dCas9-VPR cell line shows a 50-60x upregulation of MSR transcripts without dox. As these cells grow, I assume that their viability is not compromised? This would already give a hint that the effects observed here require drastic MSR overexpression. What is the PCH morphology of these cells?

With the definition of a threshold limit (new Figure 4), we can now explain why the 40-fold increased MSR transcript levels in the MSR-dCas9-Activator(-dox) samples do not induce dispersion of heterochromatic foci.

- The authors write a whole section in the Results about dCas9-VPR inducing histone hyperacetylation and recruitment of phosphorylated RNA polymerase II, without referencing

an earlier paper that has presented very similar results (see Fig. 7H and Fig. S7 in PMID 32101700 for H3K27ac and RNAPII S5P stains) (Erdel et al., 2020). I think it would be fair to do so.

We had already cited this earlier work, but have now referred to it both in the introduction and results and discussion segments.

Reviewer #3 (Remarks to the Author):

In this manuscript the authors examine the role of major satellite RNA in heterochromatin organisation and integrity. This is a question of fundamental importance, towards the understanding how heterochromatin is organised and contributes to accurate chromosomal inheritance at mitosis. Establishing an inducible epigenetic activation and repression system targeting major satellite sequences, they achieve considerable modulation of major satellite transcription in MEF cells. They go on to demonstrate that both modulations lead to alterations in heterochromatin organisation, while forced overexpression of major satellites causes chromosome segregation errors and severely impacts cell viability.

The inducible system to perturb major satellite expression with dCas9 represents an exciting technical advance and improves upon previously published systems that have modulated major satellite expression. However, the biological findings reported here are not groundbreaking, as major satellite RNA is known to be important for heterochromatin organisation both in ES cells and early mammalian embryos (e.g. Lopes Novo et al., Nat. Comm. 2022, Erdel et al., Mol. Cell 2020, Casanova et al Cell Reports 2013). The novelty of this phenotype in MEF cells represents a fairly incremental advance in knowledge. It has also been demonstrated that depletion of major satellite RNA results in chromosomal instability (Lopes Novo et al). Here such a phenotype is found after strong activation of major satellite expression, although the physiological implications of this massive overexpression are not clear (see general comment 2 below). Although the technology used here is impressive, the authors unfortunately still cannot rule out effects of recruitment of epigenetic modifiers themselves, rather than major satellite transcription itself, as causal for the phenotypes observed, in this reviewer's opinion (see general comment 3). With these and other caveats specified below in mind, I unfortunately cannot currently recommend publication of this manuscript in Nature Communications.

Major Comments

General comments

1. There is an absence throughout the manuscript of appropriate reporting of biological reproducibility, which is a foundation of experimental science. Please report the number of independent experiments performed as well as individual samples throughout.

We have carefully revised the Figures and Figure legends and now provide statistical analyses, number of independent experiments and n-numbers of individual cells or images analysed. This is now specified in the Figure legends.

2. A major concern is the leakiness of MSR-dCas9-Activator in -dox control and thus whether this can be used as an appropriate control throughout the manuscript. Although not detected by Western blot (Figure 1b), a considerable increase (50x) in MajSat RNA levels are observed in MSR-dCas9-Activator -dox conditions (Figure 1d). I question whether this condition is really 'uninduced' and thus is it an appropriate control? The authors need to compare phenotypes of MSR-dCas9-Activator -dox conditions to MSR-dCas9-Control throughout, to determine whether this 50x induction leads to changes in the phenotypes observed (e.g. Figure 2a). If not, why is 50x increase in transcription not sufficient but a 4000x fold increase in expression

over unmodified cells is necessary? For example, if a 50-fold increase in MajSat transcript levels is not sufficient to cause chromosome segregation or apoptosis, what is the physiological relevance of a 4000-fold increase?

Our new data on the definition of a threshold limit for forced MSR expression in disrupting heterochromatin (new Figure 4) (see also response to reviewer 1 and reviewer 2) addresses these questions and can explain that 40-fold increased MSR transcript levels are significantly below the threshold limit (300-fold upregulation) and do not induce dispersion of DAPI-dense foci.

In addition, we quantified mitotic defects and chromosome mis-segregation in a new supplementary Figure S7. Mitotic defects are modestly increased by 2-fold in MSR-dCas9-Activator(-dox) as compared to MSR-dCas9-Control(-dox) but are >9-fold higher in MSR-dCas9-Activator(+dox). These data support our conclusions that increasing levels of MSR RNA manifest with higher and more frequent mitotic defects. There are no cellular pathways were dysregulated between MSR-dCas9-Activator(-dox) and MSR-dCas9-Control(-dox).

3. The authors use dCas9 fused to KRAB-MeCP2 to induce repression and observe an aggregation of MajSat regions (Figure 2). However, how can the authors exclude that this effect is simply due to targeting of the KRAB-MeCP2 effectors to Major Satellite rather than via decreased levels of MajSat RNA? Similar concerns are pertinent for the Activator (VPR), which may lead to decondensation independently of transcriptional changes.

This cannot be excluded and is indeed clearly stated (“...heterochromatic regions aggregate. This could primarily be caused by increased recruitment of components of the silencing machinery and protein-based coalescence...”) in the first paragraph of the discussion.

As already explained in response to reviewer 2, approaches to uncouple RNA output from an active chromatin state has been a long-standing challenge in the field. We engaged in several new approaches, such as RNase A treatment of permeabilized cells (new supplementary Figure 6d), inhibition of RNA Pol II transcription (new Figure 5a-c) and reducing histone acetylation (new Figure 5d-g). From these new approaches, we conclude that both RNA output and a transcriptionally hyperactive chromatin state influence heterochromatin organization. These are important new data sets that we have added and also present and discuss accordingly.

Specific Comments

Figure 2

In addition to sphericity and volume in Figure 2a, number of segmented DAPI-dense regions should be reported, as the authors claim a decreased number of regions in MSR-dCas9-repressor conditions. How robust are the measurements of sphericity, volume and number of segmented MSR-foci to segmentation method. How were thresholds chosen for segmentation? If thresholds are modified are the reported effects consistent?

We did not count the number of DAPI-dense regions, but rather analysed the number of CREST puncta per DAPI-dense region, which shows higher number of CREST puncta per DAPI-dense region in the MSR-dCas9-Repressor MEF cells. These data have been moved to supplementary Figure 3a, where we now also provide statistical testing. The measurements of sphericity and volume of DAPI-dense regions are robust and maintain the observed differences when threshold levels are modified.

Figure 2b – why is the data grouped in such a way? Please represent the data in a clearer way without grouping and provide appropriate statistical tests.

The CREST data have been moved to supplementary Figure 3a, where we now also provide

statistical testing.

Figure 2c and Supplementary Figure 3 – are the presented linescans representative? Of how many images? How many independent experiments?

The linescans shown are derived from the displayed images. Images are representative from three biological replicates. The number of cells analysed are indicated in the Figure legends.

Figure 4

The authors do not mention the localisation of MajSat RNA in the control condition. Are they also associated with chromocentres? Does their MajSat RNA FISH data agree with previous research in ES cells (Lopes Novo et al). If not, why is only overexpressed MajSat associated with DAPI-rich regions? The authors should discuss the implications of this. Data should be presented also in -Dox conditions (Figure 4a).

MSR transcripts reach higher levels in mESC as compared to MEF cells. MSR RNA has been reported to largely remain associated with pericentric heterochromatin in mouse cells and is consistent with previous reports, including our work in MEF cells (Bulut-Karslioglu et al., 2012) and with Lopes Novo et al., 2022 in mESC. Signals for MSR transcripts in the MSR-dCas9-Control(+dox) are low, since exposure times for imaging were adjusted to the high signals for MSR transcripts in the MSR-dCas9-Activator(+dox). We also did the MSR RNA-FISH analyses in the uninduced conditions, where we did not observe differences for the MSR RNA signals in MSR-dCas9-Control, MSR-dCas9-Repressor and MSR-dCas9-Activator MEF cells (data not shown).

Figure 5

Chromosome segregation abnormalities shown in Figure 5c should be quantified with appropriated N numbers and statistical testing.

This is now added as new supplementary Figure S7

Figure 6

Quantification of the phenotypes referenced regarding Figure 6b should be provided.

More than 80% of MSR-dCas9-Activator MEF cells 24 hrs post-induction show early to late apoptosis, as quantified by flow cytometry in Figure 7a (right panel).

Discussion

The lack of a strong phenotype in terms of chromosomal instability upon major satellite depletion differs to a previous study in ES cells (Lopes Novo). This should be discussed.

We discuss the differences in number and size of heterochromatic foci between mouse ES cells (Lopes Novo et al., 2022), serum-free mESC and MEF cells (our study). We have extended this discussion point and now also refer to the 50% reduced level of MSR transcripts in the (Lopes Novo et al., 2022) paper.

The authors argue that 'MSR RNA output, rather than the transcription process per se, is a crucial determinant in directing the structural organization of pericentric heterochromatin'. How do they discriminate between these possibilities in this manuscript? I don't see any evidence to support this statement, except for the TALE-Onconase experiments in Supplementary Figure 1, which had modest effects.

This has indeed been the most challenging part of the study and of the revision. As explained above, we addressed this question with several new approaches, such as RNase A treatment of permeabilised cells (new supplementary Figure 6d), inhibition of RNA Pol II transcription

(new Figure 5a-c) and reducing histone acetylation (new Figure 5d-g). From these new approaches, we conclude that both RNA output and a transcriptionally hyperactive chromatin state influence heterochromatin organisation. These are important new data sets that we have added and also present and discuss accordingly.

Minor Comments

Regarding Figure 3c, reference should be made to previous research reporting similar findings (Erdel et al., Mol. Cell 2020).

We had already cited this earlier work, but have now referred to it both in the introduction and results and discussion segments.

Please provide the full sequence of MajSat gRNA.

The DNA sequence of one unit of the MSR consensus sequence is now incorporated into the diagram of Figure 1a.

Change scale of L1Md_A plot in Figure 1d

We use different scales to illustrate the significant differences in the levels of major satellite, minor satellite and LINE L1Md_A transcripts.

REVIEWER COMMENTS

Reviewer #1 (Remarks to the Author):

The authors have addressed the reviewers' comments as best they can by including a significant amount of new experimental data in Figures 4 and 5 and Figure S6. Their new data establishing the threshold level of MSR RNA expression required for disruption of heterochromatin foci in MEFs nicely explain why the relatively high basal level of MSR RNA in the MSR-dCas9-Activator cells in the absence of DOX was insufficient to cause disruption of foci, and also why the relatively small physiological increase in MSR RNA level observed in 2-cell embryos drives chromocenter formation rather than loss of foci. Their new data with RNase treatment, DRB inhibition of RNA Pol II activity, and A-485 inhibition of p300/CBP HAT activity suggest that MSR RNA expression and a transcriptionally hyperactive chromatin state are needed for the observed effects of MSR-dCas9-Activator driven MSR transcription in reducing heterochromatin condensates. Ultimately, however, more will need to be done to define the precise role of MSR RNA in the dynamics of pericentric heterochromatin foci formation and disassembly at a mechanistic level.

We thank this reviewer for the balanced and positive evaluation of our revised manuscript.

Reviewer #2 (Remarks to the Author):

The authors have thoroughly revised and strengthened their manuscript entitled "Forced expression of MSR repeat transcripts above a threshold limit breaks heterochromatin organisation". They have established a (relative) threshold of MSR transcripts above which heterochromatin foci are dispersed, and they have conducted additional experiments to test the influence of MSR transcription and chromatin state on heterochromatin organization. I have some mostly minor comments:

We thank this reviewer for the balanced and positive evaluation of our revised manuscript.

1. The authors conclude that both RNA output and a transcriptionally hyperactive chromatin state influence heterochromatin organization. I wonder if the title of the manuscript could be adjusted to convey this conclusion more clearly.

We opt to maintain the title as it directly specifies the experimental approach. The conclusion that both RNA output and a transcriptionally hyperactive chromatin state influence heterochromatin organisation is clearly stated in the abstract.

2. The authors determine a threshold level of MSR transcripts above which heterochromatin is dispersed. According to Fig. 4d, mammary gland cells have higher MSR transcript levels (50,000 normalized reads) than 6KO cells (20,000 normalized reads) that have dispersed heterochromatin. Presumably mammary gland cells contain focal heterochromatin structures and do not have the problems associated with dispersed heterochromatin that the authors describe here, which speaks against an absolute threshold. I suppose that the authors indicate relative thresholds (300-fold instead of a number of normalized counts) for this reason. Could the authors speculate how the cell type-dependence of this threshold can be explained? Are there any obvious changes in heterochromatin composition in mammary gland cells?

We have used a proportional conversion of RT-qPCR MSR expression levels to also define a threshold limit for the RNA-seq reads in wt MEF cells, Suv39h double-null, 6KO and MSR-dCas9-Activator (+/- dox) MEF cells and in the mammary gland cells (adapted Figure 4d).

This now shows that MSR RNA-seq reads in the 6KO MEF cells reach the threshold limit to disperse heterochromatin and that the MSR RNA-seq reads in the mammary gland cells (both non-tumour and tumour) are even above the threshold limit. Unfortunately, the structural organisation of heterochromatin in non-tumour vs. tumour mammary gland cells had not been analysed in these studies. Further, it is known that mESC have a less pronounced focal organisation of heterochromatin than MEF cells and heterochromatin organisation may also vary in other cell types, particularly if they differ in the amount or composition of heterochromatin components. We thank this reviewer for this insightful comment and have clarified a possible cell-type dependence for a threshold limit in the discussion.

3. I find it difficult to reconcile the discussion about RNA-HP1 condensates with the authors' data and the data in ref. 10. An increase of MSR transcript could indeed saturate HP1 RNA binding and disfavor inter-nucleosomal protein-protein interactions, explaining the switch from aggregated to dispersed heterochromatin. However, Suv39h double-null MEFs where most HP1 is displaced from heterochromatin are fairly similar to wildtype MEFs: They have focal heterochromatin structures that become dispersed when dCas9-VPR is recruited (ref 10). This seems to suggest that HP1 does not have a strong effect on the aggregation of heterochromatin foci and their VPR-dependent dispersion.

In the discussion, we had referred to an HP1-driven model that connects RNA binding and RNA-HP1 condensates with a heterochromatin scaffold. The data from Suv39h double-null MEF cells published by us (Peters et al., 2001) and their more recent analysis in ref. 10 (Erdel et al., 2020) indicate also HP1-independent mechanisms to contribute to a heterochromatin scaffold. We have added this in the revised discussion.

Reviewer #3 (Remarks to the Author):

While the authors have performed a significant amount of work, with the addition of Figures 4 and 5, unfortunately I am still not in support of publication as my major comments have not been fully addressed.

While reviewer three also acknowledges our efforts in having added new data sets, this reviewer remains unsupportive. However, most of the criticism is on absence of statistical testing and lack of numbering independent experiments, which is then taken as an argument to question the robustness of our data. We strongly disagree with this exaggerated criticism, as we have a long-standing record in generating high-quality and reproducible data sets.

Major Comments

General comments

1. There is an absence throughout the manuscript of appropriate reporting of biological reproducibility, which is a foundation of experimental science. Please report the number of independent experiments performed as well as individual samples throughout.

We have carefully revised the Figures and Figure legends and now provide statistical analyses, number of independent experiments and n-numbers of individual cells or images analysed. This is now specified in the Figure legends.

This point is still not addressed. There is still a lack of reporting of number of independent experiments throughout the manuscript. For example, Figures 1b and c do not mention the number of times the experiment was repeated. This is also not addressed in the examples presented in the specific comments below.

We have expanded the Methods section to indicate how many times experiments were

repeated, also for the Western blot analyses (Figure 1b, Figure 4a and supplementary Figure 6a) or for the imaging data.

2. A major concern is the leakiness of MSR-dCas9-Activator in -dox control and thus whether this can be used as an appropriate control throughout the manuscript. Although not detected by Western blot, a considerable increase (50x) in MajSat RNA levels are observed in MSR-dCas9-Activator -dox conditions (Figure 1d). I question whether this condition is really 'uninduced' and thus is it an appropriate control? The authors need to compare phenotypes of MSR-dCas9-Activator -dox conditions to MSR-dCas9-Control throughout, to determine whether this 50x induction leads to changes in the phenotypes observed (e.g. Figure 2a). If not, why is 50x increase in transcription not sufficient but a 4000x fold increase in expression over unmodified cells is necessary? For example, if a 50-fold increase in MajSat transcript levels is not sufficient to cause chromosome segregation or apoptosis, what is the physiological relevance of a 4000-fold increase?

Our new data on the definition of a threshold limit for forced MSR expression in disrupting heterochromatin (new Figure 4) (see also response to reviewer 1 and reviewer 2) addresses these questions and can explain that 40-fold increased MSR transcript levels are significantly below the threshold limit (300-fold upregulation) and do not induce dispersion of DAPI-dense foci. In addition, we quantified mitotic defects and chromosome mis-segregation in a new supplementary Figure S7. Mitotic defects are modestly increased by 2-fold in MSR-dCas9-Activator(-dox) as compared to MSR-dCas9-Control(-dox) but are >9-fold higher in MSR-dCas9-Activator(+dox). These data support our conclusions that increasing levels of MSR RNA manifest with higher and more frequent mitotic defects. There are no cellular pathways dysregulated between MSR-dCas9-Activator(-dox) and MSR-dCas9-Control(-dox).

I applaud the authors for attempting to address a threshold limit of MSR transcripts for heterochromatin integrity, in a new Figure 4, which does address the leakiness concern of the uninduced condition. However, the following points limit my enthusiasm for this.

- In Figure 4a (and Supplementary Figure 6a) the number of experiments is not reported.

This has now been done in the expanded Methods section for the Western blot analyses.

- Statistical testing is not shown for Figure 4c (and Supplementary Figure 6c). Thus, the conclusion that dispersion is observed at 2 ng/ml dox for example, is not supported. As this is the basis for setting the threshold of MSR transcripts, I am not in support of publication of this conclusion.

The imaging data in Figure 4c and supplementary Figure 6c were done in three independent experiments and with $n > 270$ cells per sample (Figure 4c) or $n > 170$ cells per sample (Supplementary Figure 6c). This is indicated in the Figure legends. We therefore consider these data as robust.

- In many new panels they perform a counting of dispersed vs undispersed heterochromatin phenotypes. As far as I can tell the determination of how dispersed vs undispersed were defined and how was the data binarized in such a way? Was this done manually? If so, was this done blind? How was blinding achieved? Quantification of dispersion should be provided, similarly to Figure 2, with volume and number of domains. This also applies to Supplementary Figure 6d and Figure 5c/f.

The imaging analysis of undispersed vs. dispersed heterochromatin has now been detailed in the Methods section. For each sample in Figure 5c and Figure 5f, $n > 135$ cells were analysed and quantification is provided by the shown stacked bar graphs. We therefore

consider these data as robust.

- I do not follow the calculations for fold change in MSR RNA sequences (lines 323-327). They describe 220,000 normalized reads in the induced condition, which compared to 600 normalized reads in uninduced control, represents a 367-fold increase, not >470-fold as they report. Similarly, compared to 2000 normalized reads in the uninduced MSR-dCas9-Activator condition, my calculation gives a 110-fold increase not >150-fold as they describe.

We have removed the numbers for fold increase of MSR RNA-seq reads and instead describe a threshold limit also for MSR RNA-seq reads that has been calculated through a proportional conversion of the RT-qPCR MSR expression levels.

- If a threshold of 300-fold MSR upregulation is necessary for dispersion of heterochromatin, why is a complete collapse observed in 6KO MEF cells, where only an 80-fold MSR upregulation is observed compared to wt cells?

The definition of a 300-fold threshold for MSR derepression shown in Figure 4b is derived from RT-qPCR analyses. RT-qPCR to quantify differences in MSR transcript levels is likely to overestimate the relative abundance of repeat transcripts since there are multiple target sites for PCR primers in multi-copy repeat RNA. Thus, RT-qPCR expression levels cannot be directly compared to the number of normalized read counts detected by RNA-seq. We used a proportional conversion of the RT-qPCR MSR expression levels (shown in Figure 4b) to calculate a threshold limit also for the MSR RNA-seq reads (adapted Figure 4d). This now shows that MSR RNA-seq reads in the 6KO MEF cells reach the threshold limit to disrupt heterochromatin.

- How specific is this threshold for MEF cells? It is not clear how this threshold would apply to other cell types. A cell-type threshold dependence could exist for a number of reasons, including different organisation and proteome of chromocenter-associated heterochromatin in different cell types and different recruitment of activators. At least, the authors should specify that this specific threshold would apply to MEF cells and cannot necessarily crossover to other cell types. Thus, the discussion of the physiological relevance and the direct application of this threshold to other cell types in the relevant section in the discussion should be toned down.

This is a valid point and heterochromatin organisation may indeed vary in other cell types, particularly if those differ in the amount or composition of heterochromatin components. We have clarified in the discussion that the 300-fold threshold limit has been defined in MEF cells and that it may fluctuate in other cell types and under variable compositions of heterochromatin.

3. The authors use dCas9 fused to KRAB-MeCP2 to induce repression and observe an aggregation of MajSat regions (Figure 2). However, how can the authors exclude that this effect is due simply to targeting of the KRAB-MeCP2 effectors to Major Satellite rather than via decreased level of MajSat RNA? Similar concerns are pertinent for the Activator (VPR), which may lead to decondensation independently of transcriptional changes.

This cannot be excluded and is indeed clearly stated (“...heterochromatic regions aggregate. This could primarily be caused by increased recruitment of components of the silencing machinery and protein-based coalescence...”) in the first paragraph of the discussion. As already explained in response to reviewer 2, approaches to uncouple RNA output from an active chromatin state has been a long-standing challenge in the field. We engaged in several new approaches, such as RNase A treatment of permeabilized cells (new supplementary Figure 6d), inhibition of RNA Pol II transcription (new Figure 5a-c) and reducing histone acetylation (new Figure 5d-g). From these new approaches, we conclude

that both RNA output and a transcriptionally hyperactive chromatin state influence heterochromatin organization. These are important new data sets that we have added and also present and discuss accordingly.

Apologies for missing this point in the discussion previously. I commend the authors for attempting to address this difficult point using three different approaches. However, I have a number of concerns about the data:

- The quantification of dispersed vs undispersed in Figure 5c/f is not clearly described and statistical testing is not shown (see 2nd and 3rd points above).

The imaging analysis of undispersed vs. dispersed heterochromatin has now been detailed in the Methods section. For each sample in Figure 5c and Figure 5f, $n > 135$ cells were analysed and quantification is provided by the shown stacked bar graphs. We therefore consider these data as robust.

- The results in Figures 5f and 5g appear inconsistent. In Figure 5f a 5 μ M concentration of A-485 leads to 30% dispersion. In Figure 5g, under comparable conditions 75% dispersion is observed (without DRB). The only difference as far as I can tell is that Dox is applied for only 1h in 5g compared to 6h in 5f. However, this should decrease the amount of MSR transcript produced, and thus fewer dispersed chromocentres should be observed.

Figure 5f and Figure 5g describe two different experiments with either a 6 hrs or a 1 hrs induction of the MSR-dCas9-Activator. Each experiment was repeated three times and the data are robust as they are shown in the Figures.

- The results of these experiments are not conclusive, even ignoring the caveats described above. Unfortunately, the acetyltransferase inhibitor leads to a decrease in transcription of MSR and thus an effect of acetylation independently from MSR transcripts cannot be ascribed. The transcriptional inhibition or RNaseA treatment do not rescue heterochromatin dispersion (although it is not clear if either of these differences are reproducible or statistically significant), suggesting that transcription or the RNA are not involved in the dispersion. The only evidence that the MSR transcript itself may be involved in the dispersion is in Figure 5g, which shows a difference +/- DRB only at 5 μ M A-485, the results of which are not consistent with Figure 5f, as mentioned above. Thus, it is still not clear to what extent the MSR transcripts vs the recruitment of activators contributes to heterochromatin integrity.

We strongly disagree with this exaggerated criticism and rebut that our data are questioned as not being reproducible or statistically significant. We have engaged in three different approaches (RNase A incubation, RNAPII inhibition and HAT inhibition) to address whether RNA output could possibly be uncoupled from a transcriptionally activated chromatin state. We have clearly described the results from these three approaches and conclude that the structural organisation of heterochromatin is governed by both chromatin-associated MSR RNA and the transcriptional chromatin state. We consider this insight as an important advance and future studies are required to fully dissect the function of MSR RNA independent of an altered transcriptional chromatin state. The challenges for these studies were already pointed out in the discussion of our revised manuscript.

Specific Comments

Figure 2

In addition to Sphericity and Volume in Figure 2a, number of segmented DAPI-dense regions should be reported, as the authors claim a decreased number of regions in MSR-dCas9-repressor conditions.

How robust are the measurements of sphericity, volume and number of segmented MSR-foci to segmentation method. How were thresholds chosen for segmentation? If thresholds are modified are the reported effects consistent?

We did not count the number of DAPI-dense regions, but rather analysed the number of CREST puncta per DAPI-dense region, which shows higher number of CREST puncta per DAPI-dense region in the MSR-dCas9-Repressor MEF cells. These data have been moved to supplementary Figure 3a, where we now also provide statistical testing. The measurements of sphericity and volume of DAPI-dense regions are robust and maintain the observed differences when threshold levels are modified.

In lines 188-189 the authors still claim that heterochromatic regions are reduced in number in MSR-Cas9-Repressor conditions. The data is still not in the manuscript to support this statement.

We have removed the statement on reduced number of heterochromatic regions but maintain that heterochromatin aggregates upon induction of the MSR-dCas9-Repressor.

Figure 2b – why is the data grouped in such a way? Please represent the data in a clearer way without grouping and provide appropriate statistical tests.

The CREST data have been moved to supplementary Figure 3a, where we now also provide statistical testing.

The data is now presented more clearly and the number of quantified regions are described. However, they still do not describe the number of independent experiments.

The number of independent experiments (n=3) is now given in the legend.

Figure 2c and Supplementary Figure 3 – are the presented linescans representative? Of how many images? How many independent experiments?

The linescans shown are derived from the displayed images. Images are representative from three biological replicates. The number of cells analysed are indicated in the Figure legends.

The number of independent experiments is still not reported in the manuscript.

The number of independent experiments for image analysis is now provided in the Methods section.

Figure 4

The authors do not mention the localisation of MajSat RNA in the control condition. Are they also associated with chromocentres? Does their MajSat RNA FISH data agree with previous research in ES cells (Lopes Novo et al).

Data should be presented also in -Dox conditions (Figure 4a).

If not, why is only overexpressed MajSat associated with DAPI-rich regions? The authors should discuss the implications of this.

MSR transcripts reach higher levels in mESC as compared to MEF cells. MSR RNA has been reported to largely remain associated with pericentric heterochromatin in mouse cells and is consistent with previous reports, including our work in MEF cells (Bulut-Karslioglu et al., 2012) and with Lopes Novo et al., 2022 in mESC. Signals for MSR transcripts in the MSR-dCas9- Control(+dox) are low, since exposure times for imaging were adjusted to the high signals for MSR transcripts in the MSR-dCas9-Activator(+dox). We also did the MSR RNA-FISH analyses in the uninduced conditions, where we did not observe differences for

the MSR RNA signals in MSR-dCas9-Control, MSR-dCas9-Repressor and MSR-dCas9-Activator MEF cells (data not shown).

The faint MSR AS and S signal in control conditions is not localized to pericentric heterochromatin, as they claim above. There appears to be low level nuclear signal in both conditions, that is absent in the Cas9-Repressor conditions, suggesting that this faint signal is specific. This point is not sufficiently addressed.

This interpretation is not correct. The faint MSR AS and S signals in the control conditions are largely overlapping with DAPI foci and with Cas9-positive regions and are consistent with previously published data (Bulut-Karslioglu et al., 2012).

Figure 5

Chromosome segregation abnormalities shown in Figure 5c should be quantified with appropriated N numbers and statistical testing.

This is now added as new supplementary Figure S7

The quantification has now been added. However, the number of independent experiments is still not reported in the manuscript.

The number of independent experiments (n=2) is now given in the legend.

Figure 6

Quantification to the phenotypes referenced in Figure 6b should be provided.

More than 80% of MSR-dCas9-Activator MEF cells 24 hrs post-induction show early to late apoptosis, as quantified by flow cytometry in Figure 7a (right panel).

Number of independent experiments are not described.

The number of independent experiments (n=3) is now given in the Methods.

Discussion

The lack of a strong phenotype in terms of chromosomal instability upon major satellite depletion differs to a previous study in ES cells (Lopes Novo). This should be discussed.

We discuss the differences in number and size of heterochromatic foci between mouse ES cells (Lopes Novo et al., 2022), serum-free mESC and MEF cells (our study). We have extended this discussion point and now also refer to the 50% reduced level of MSR transcripts in the (Lopes Novo et al., 2022) paper.

This has been addressed.

The authors argue that 'MSR RNA output, rather than the transcription process per se, is a crucial determinant in directing the structural organization of pericentric heterochromatin'. How do they discriminate between these possibilities in this manuscript? I don't see any evidence to support this statement, except for the TALE-Onconase experiments in Supplementary Figure 1, which had modest effects.

This has indeed been the most challenging part of the study and of the revision. As explained above, we addressed this question with several new approaches, such as RNase A treatment of permeabilised cells (new supplementary Figure 6d), inhibition of RNA Pol II transcription (new Figure 5a-c) and reducing histone acetylation (new Figure 5d-g). From these new approaches, we conclude that both RNA output and a transcriptionally

hyperactive chromatin state influence heterochromatin organisation. These are important new data sets that we have added and also present and discuss accordingly.

See my response to major comment 3.

We had thoroughly revised the original manuscript and maximised our efforts to address this challenging question as best as we can. We can now provide conclusive new insight that the structural organisation of heterochromatin is governed by both chromatin-associated MSR RNA and the transcriptional chromatin state. We have now reached our limits in responding to this reviewer and future studies are required to fully dissect the function of MSR RNA independent of an altered transcriptional chromatin state.

REVIEWER COMMENTS

Reviewer #2 (Remarks to the Author):

The authors have replied to my questions in the last revision round and I am fine with their answers.

Overall, it is a bit unfortunate that it remains unclear if the contribution of the MSR transcripts themselves or that of the active chromatin state is stronger in breaking heterochromatin. As I had suggested in the first revision round, I thought that this could have been tested by directly tethering the MSR transcripts (produced elsewhere) instead of recruiting the transcriptional Activator (to transcribe endogenous MSRs). In any case, I agree with the authors that their data suggest that both the MSR transcripts and the active chromatin state affect heterochromatin organisation. This model suggests that there is a threshold limit for MSR transcripts, which receives lots of attention in the manuscript, but also a threshold limit for active histone marks (or for components of the transcription machinery that accumulate over MSRs), above which heterochromatin organisation is disrupted. This aspect could be presented in a more balanced way, at least in the Discussion. The existence of another threshold, one for active histone marks and activator proteins, could also explain that the threshold for MSR transcripts is cell type-dependent, as it is not the only determinant of heterochromatin organisation.

We thank this referee for the balanced insight and have added this valid point in the discussion (lines 539ff): *“As with an MSR RNA threshold limit, there may also be a threshold limit for histone acetylation and associated transcriptional activators above which heterochromatin organisation becomes disrupted”*.

Reviewer #3 (Remarks to the Author):

The authors argue based on reputation that their data is robust: 'We strongly disagree with this exaggerated criticism, as we have a long-standing record in generating high-quality and reproducible data sets.' This is unfounded as science is assessed by individual merit rather than reputation. As many of my relatively simple requests were not addressed in the previous round of review (even though the authors claimed they were) I naturally requested them again and thus I do not believe my criticism is exaggerated. Many of my points have now been addressed, but I still have some outstanding points, most importantly pertaining to the definition of the 300-fold threshold, an important conclusion of the manuscript.

1. I thank the authors for now providing reproducibility reporting information.
2. The experimental variability (error bars across independent experiments) and statistical testing are still not shown for Figure 4c and Supplementary Figure 6c. Thus, how can they conclude that 2ng/ml dox (which equates to 300-fold induction) induces dispersion? The apparent increase in cells with dispersed heterochromatin observed at 2ng/ml could be within experimental variability. Moreover, I am still not convinced by the manual quantification of dispersed vs undispersed foci. No blinding is described, and thus the authors cannot exclude subconscious bias in this counting. As far as I can tell a more unbiased quantification of foci number and volume is not provided, as requested.

We have repeated the experiments that define the threshold limit of 300-fold deregulation of MSR transcripts. This was done with single blind counting from three independent experiments and indicates < 5% experimental variability and excludes unconscious bias. The data are added as updated Figure 4c (totaling n > 240 cells) and updated supplementary Figure 6c (totaling n > 150 cells), including statistical testing in Figure and Figure legends. Importantly, this validation confirms the threshold limit of 300-fold deregulation of MSR

transcripts. The single blind testing and quantification of undispersed vs. dispersed DAPI-dense foci is now also described in the Methods section (lines 732ff).

3. No explanation is provided for the inconsistent results between Figure 5f and g.

We have repeated the HAT inhibition of Figure 5g. Compared to the previous result, there is only modest to no difference in the quantification of the data. Previous quantification for the A-485 titration was 91%, 84%, 75% and 36% of dispersed DAPI-dense foci. Repeat quantification is 91%, 85%, 63% and 44% of dispersed DAPI-dense foci (now shown as updated Figure 5g). Although the experiments in Figure 5f and Figure 5g differ by varying the induction times for MSR-dCas9-Activator (6hrs in Figure 5f and 1hrs in Figure 5g), both start with comparable percentages of dispersed DAPI-dense foci at 0 μ M and 1 μ M A-485 (without DRB, top row of Figure 5g). At 5 – 10 μ m A-485, percentages of undispersed DAPI-dense foci are lower (37-56%) in Figure 5g (top row, without DRB) as compared to Figure 5f (70-84%). We do not consider this as inconsistent but to reflect two independent experimental conditions that both demonstrate HAT inhibition to prevent dispersion of focal heterochromatin upon induction of the MSR-dCas9-Activator.

Specific comments

Figure 4

The authors still claim that the signal of the MSR RNA FISH in control conditions overlaps with DAPI and Cas9 foci. This is clearly not the case, especially in the image shown with the ASR probe.

In the revised manuscript, we do not claim and are not making a statement on the localisation of MSR RNA FISH signals in MSR-dCas9-Control MEF cells (lines 269ff).

REVIEWER COMMENTS

Reviewer #2 (Remarks to the Author)

The authors have addressed my remaining point in the Discussion and I support publication.

We thank the reviewer for their comments and support of the manuscript.

Reviewer #3 (Remarks to the Author)

I thank the authors for adding the statistical testing to Figure 4c. Please could the authors also add error bars as requested or individual data points to show the experimental variability. This also applies to similar quantification in Figures 5c and f and Supplementary Figure 6c.

We have added error bars to Figure 4c. In addition, we have applied similar quantification to Figures 5c, f and Supplementary Figure 6c and have included error bars.

All other points have been addressed.